# Multi-target mode of action of silver against *Staphylococcus aureus* endows it with capability to combat antibiotic resistance

Haibo Wang[1], Minji Wang[2,3], Xiaohan Xu[1], Peng Gao[4], Zeling Xu[2], Qi Zhang[1], Hongyan Li[1], Aixin Yan[2], Richard Yi-Tsun Kao[4] & Hongzhe Sun[1]✉

The rapid emergence of drug resistant *Staphylococcus aureus* (*S. aureus*) poses a serious threat to public health globally. Silver (Ag)-based antimicrobials are promising to combat antibiotic resistant *S. aureus*, yet their molecular targets are largely elusive. Herein, we separate and identify 38 authentic $Ag^+$-binding proteins in *S. aureus* at the whole-cell scale. We then capture the molecular snapshot on the dynamic action of $Ag^+$ against *S. aureus* and further validate that $Ag^+$ could inhibit a key target 6-phosphogluconate dehydrogenase through binding to catalytic His185 by X-ray crystallography. Significantly, the multi-target mode of action of $Ag^+$ (and nanosilver) endows its sustainable antimicrobial efficacy, leading to enhanced efficacy of conventional antibiotics and resensitization of MRSA to antibiotics. Our study resolves the long-standing question of the molecular targets of silver in *S. aureus* and offers insights into the sustainable bacterial susceptibility of silver, providing a potential approach for combating antimicrobial resistance.

[1] Department of Chemistry, State Key Laboratory of Synthetic Chemistry, CAS-HKU Joint Laboratory of Metallomics on Health and Environment, The University of Hong Kong, Hong Kong S.A.R., People's Republic of China. [2] School of Biological Sciences, The University of Hong Kong, Hong Kong S.A.R., People's Republic of China. [3] School of Chemistry and Molecular Engineering, East China Normal University, Shanghai, People's Republic of China. [4] Department of Microbiology, The University of Hong Kong, Hong Kong S.A.R., People's Republic of China. ✉email: hsun@hku.hk

Both hospital-acquired and community-acquired staphylococcal infections are increasing steadily with *Staphylococcus aureus* (*S. aureus*) being one of the five most common causes of hospital-acquired infections[1]. *S. aureus* is the causative agent of a variety of diseases, such as skin infection, food poisoning, bone/joint infection, and bacteremia, ranging from subacute superficial skin infection to life-threatening septicemia[2,3]. The rise in incidence has been accompanied by an increase in antibiotic-resistant strains, notably methicillin-resistant *S. aureus* (MRSA)[4,5] and vancomycin-resistant strains[6]. Moreover, the outbreak of Coronavirus Disease 2019 (COVID-19) pandemic may further increase antimicrobial resistance due to the heavy use of antibiotics to treat patients infected with Severe Acute Respiratory Syndrome Coronavirus 2 (SARS-COV-2)[7,8]. Given the rapid emergence of drug-resistant *S. aureus* but a lack of antibiotic-development pipeline, alternative strategies are urgently needed to combat antibiotic-resistant *S. aureus*.

Metal ions have been historically used as antimicrobial agents owing to their inherent broad-spectrum antimicrobial properties and less chance of resistance[9–14]. There is a growing interest in revitalizing metal-based compounds as promising alternatives to tackle antimicrobial resistance crisis[9,13,15–18]. For instance, a gallium-based drug (Ganite®) has been shown to be effective in the treatment of chronic *Pseudomonas aeruginosa* airway infections both in mouse infection models and in a Phase I clinical trial in individuals with cystic fibrosis, and exhibits low rates in the development of resistance comparing to antibiotics[19]. Recently, bismuth compounds have also been uncovered as resistance breakers to restore antibiotic efficacy[20].

Silver ions ($Ag^+$) and silver nanoparticles (AgNPs) have been used as antimicrobial agents for centuries and are still being widely used in healthcare and food industry[9,10,21–25]. Despite the enormous research efforts, the antimicrobial mechanisms of $Ag^+$ against *S. aureus* remain largely obscure. Up to now, most studies focus on the up- and down-regulated proteins in *S. aureus* post exposure to $Ag^+$ or AgNPs by quantitative proteomic approaches[26,27]. However, no proteins that directly bind $Ag^+$ have been identified in *S. aureus* systematically owing to the complexity of metal–protein interactions in cells and the chemistry of $Ag^+$ as well as a lack of appropriate techniques, which hinders in-depth understanding the mode of action of antibacterial silver[28–30]. Several approaches, e.g., laser ablation inductively coupled plasma mass spectrometry (LA-ICP-MS)[31], liquid chromatography hyphenated with ICP-MS[32,33], synchrotron X-ray fluorescence spectrometry (SXFS)[34], and fluorescence labeling[19,35], have been developed to mine unknown metalloproteomes in microbes. However, these methods are limited by either separating resolution, detection sensitivity, or accessibility of the facility, which hinders extensive exploration of silver directly targeted proteins in bacteria. Previously, we have developed a robust approach, i.e., LC-GE-ICP-MS, through combining liquid chromatography (LC) with gel electrophoresis (GE) hyphenated with inductively coupled plasma and mass spectroscopy (ICP-MS)[36,37], which was successfully applied to uncover $Ag^+$-bound proteins in *Escherichia coli*[38].

Herein, we identify $Ag^+$-proteome in *S. aureus* and delineate the dynamic bactericidal mode of action of antibacterial $Ag^+$ against *S. aureus* at the molecular level. We show that silver exerts bactericidal activity through targeting multiple biological pathways via functional disruption of key proteins. Such a shotgun mode of action of $Ag^+$ endues the sustainable antimicrobial effect of silver. Importantly, we have shown that silver could be further exploited to enhance the efficacy of conventional antibiotics as well as to resensitize MRSA to antibiotics.

## Results

**Mining $Ag^+$-binding proteins in *S. aureus*.** We first measured the minimal inhibitory concentration (MIC)[39] of $AgNO_3$ against *S. aureus Newman* in brain heart infusion (BHI) medium and the $MIC_{50}$ of 20 μg/mL (118 μM, with effective concentration of 45 μM) was used to treat *S. aureus* for protein identification and other physiological tests (Fig. 1a). The addition of 20 μg/mL $AgNO_3$ led to a longer lag phase ($\lambda = 2.79$ h) and a decrease of maximum specific growth rate ($\mu_{max} = 0.09$ OD/h) of *S. aureus* when comparing with these of untreated cells ($\lambda = 0.90$ h and $\mu_{max} = 0.24$ OD/h) (Fig. 1b). The time-dependent uptake study shows that the intracellular concentration of $Ag^+$ increased up to 2 h of exposure (Fig. 1c). The accumulated $Ag^+$ was found mainly in proteins, with 30–40% and 20–30% in soluble and membrane proteins, respectively (Fig. 1d), indicating that proteins are the major targets of $Ag^+$.

To explore $Ag^+$-binding proteins, *S. aureus* cells after exposure to 20 μg/mL $AgNO_3$ for 0, 5, 30, and 60 min were harvested and subjected to protein extraction. The $Ag^+$-binding proteins were first separated and monitored by the 1D GE-ICP-MS. Same amounts of extracted protein samples were added for each 1D GE-ICP-MS analysis with $^{127}I$-labeled proteins as internal standards to calibrate the molecular weights (MWs) and signal intensity (Supplementary Fig. 1a). The time-dependent profiles of both soluble (Supplementary Fig. 1b–d) and membrane (Supplementary Fig. 1e–g) proteins showed constantly increasing signals of $^{107}Ag$ within 1 h but the MWs of the peaks are similar. The majority $Ag^+$-binding proteins have the MW ranging from 10 to 50 kDa and 10 to 30 kDa for the soluble and membrane proteins, respectively. However, overlaps of peaks were observed in 1D GE-ICP-MS profiles, indicating that additional separation is needed to resolve the overlapped peaks.

We then applied our customized approach LC-GE-ICP-MS[38] to explore $Ag^+$-binding proteins in *S. aureus*. First dimensional separation with LC showed that the majority soluble proteins have isoelectric points (pIs) in the range of 4.0–6.0 (Supplementary Fig. 2). The collected fractions from LC were further separated with GE and silver signals were monitored by ICP-MS. As depicted in Fig. 1e and Supplementary Fig. 3a. In all, 31 $Ag^+$-associated proteins from the soluble fraction were identified, with majority of $Ag^+$-binding proteins in *S. aureus* having MWs less than 50 kDa and pIs less than 6. The corresponding peaks were fractionized, collected, and subjected to MALDI-TOF-MS identification through peptide mass fingerprinting (Supplementary Tables 15–52). As the first dimensional separation is based on LC, proteins with similar or identical MWs but different pIs were well separated by LC-GE-ICP-MS, such as AhpD (MW 16.6 kDa, pI 5.39) and RplI (MW 16.3 kDa, pI 9.28), TnP (MW 20.2 kDa, pI 6.43) and FtnA (MW 19.5 kDa, pI 4.68), and CcpA (MW 36.1 kDa, pI 5.46) and PdhB (MW 36.7 kDa, pI 6.47) (Supplementary Table 1). Certainly, proteins with similar pIs but different MWs were also well separated by GE. For the membrane proteins, seven $Ag^+$-binding proteins were resolved by 1D GE-ICP-MS (Supplementary Fig. 1g and Supplementary Table 2), including Fib, DM13, RplF, RpsC, RpsB, AtpA, and Map. Among the 38 $Ag^+$-binding proteins, four proteins show relatively high $^{107}Ag$ intensity, i.e., RplI, TrxA, TrxB, and PflB (Fig. 1f).

To verify whether LC-GE-ICP-MS could precisely track $Ag^+$-binding proteins in *S. aureus*, we overexpressed and purified three proteins (RpoA, Pgl, and Gnd) (Supplementary Tables 3–5), then incubated them with $Ag^+$ prior to analysis of their $Ag^+$-binding capability by GE-ICP-MS. We observed single $^{107}Ag$ peaks in each GE-ICP-MS profile at the MW ca. 35, 39, and 50 kDa, corresponding to the monomers of $Ag^+$-bound RpoA, Pgl, and Gnd (Fig. 1g). In contrast, no

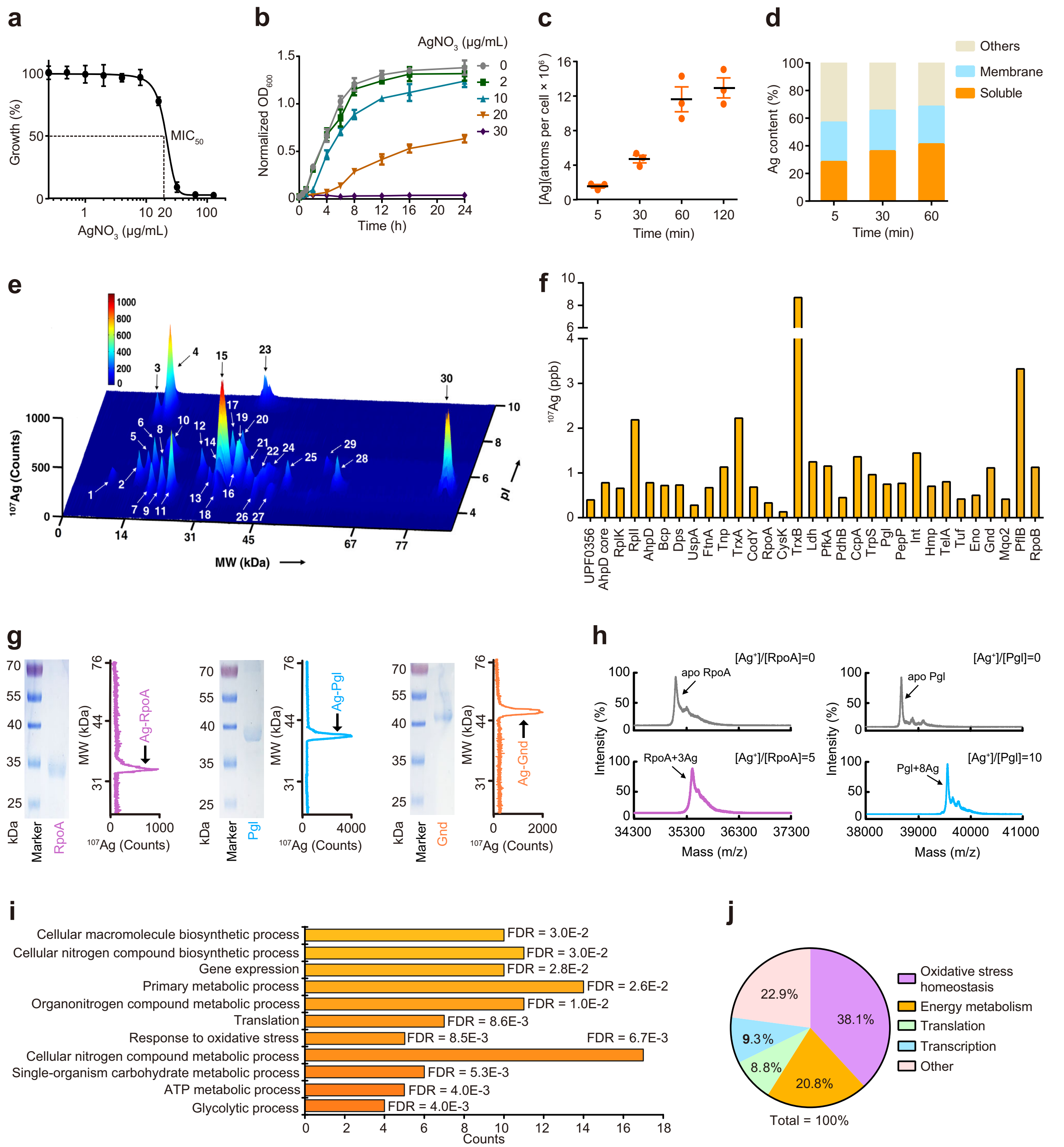


$^{107}$Ag peak in GE-ICP-MS profile at the MW ca. 30 kDa was observed (Supplementary Fig. 3b) when $Ag^+$ was incubated with glutamyl endopeptidase (GEP) (~29 kDa), a protein that is not identified by LC-GE-ICP-MS. We further examined the binding stoichiometry of $Ag^+$ to RpoA and Pgl by MALDI-TOF MS. In the absence of $Ag^+$, the peaks at $m/z$ of 35100.3 and 38682.9 were observed, corresponding to RpoA and Pgl monomers, respectively (Fig. 1h). New peaks at $m/z$ of 35421.6 and 39542.4 appeared after incubation the proteins with $Ag^+$, corresponding to RpoA and Pgl with three and eight $Ag^+$ ions bound, respectively (*calcd* $m/z$ of 35423.9, and 39545.8) (Fig. 1h). The binding stoichiometries were further confirmed by the isothermal titration calorimetry experiments, which also gives rise to dissociation constants in the range of 0.53–9.71 μM between $Ag^+$ and these three proteins (Supplementary Fig. 4), generally in agreement with our previous studies (at sub-micro molar level)[36,40]. Moreover, among the identified $Ag^+$-binding proteins, CcpA[41], TrxA[42], TrxB[42], and Ldh[43] were reported as $Ag^+$-binding protein targets previously. These results collectively corroborated that LC-GE-ICP-MS uncovers $Ag^+$-associated proteins accurately and enables the authentic $Ag^+$-binding proteins to be tracked in *S. aureus*.

**Bioinformatics analysis**. We next explored the biological insights of the identified $Ag^+$-binding proteins by STRING[44], a widely used search tool for functional enrichment analysis based on a

**Fig. 1 Mapping $Ag^+$-proteome in *S. aureus*.** **a** Minimum inhibitory concentration (MIC) test of $AgNO_3$ against *S. aureus* in BHI culture medium. The measured $MIC_{50}$ of 20 μg/mL (118 μM) $AgNO_3$ was applied to treat *S. aureus* for extraction of proteins ($n = 3$). The $OD_{600}$ of *S. aureus* cells without treatment of $AgNO_3$ was set as 100%. The growth of *S. aureus* treated with different concentrations of $AgNO_3$ was normalized to that of the control group. **b** Growth curves of *S. aureus* after treatment of different concentrations of $AgNO_3$ ($n = 3$). **c** Time-dependent uptake of $Ag^+$ in *S. aureus* cells after treatment with 20 μg/mL $AgNO_3$ for 5, 30, 60, and 120 min ($n = 3$). **d** Distribution of $Ag^+$ in different fractions of *S. aureus* after treatment with 20 μg/mL $AgNO_3$ for 5, 30, and 60 min ($n = 3$). The other stands for the pellet that could not be dissolved by 1% SDS, which is mainly composed of the cell wall. **e** Exploration of $Ag^+$-binding proteins in soluble fraction of *S. aureus* with LC-GE-ICP-MS ($n = 3$). 31 $Ag^+$-associated proteins were observed, separated, and identified. In the 3D figure, the *x*-axis, *y*-axis, and *z*-axis show the isoelectric points of $Ag^+$-binding proteins according to the LC separation, the molecular weights that were referenced to the $^{127}I$-labeled protein markers, and the intensity (counts) of Ag contained in proteins, respectively. 1 (UPF0356), 2 (AhpD core domain), 3 (RplK), 4 (RplI), 5 (AhpD), 6 (Bcp), 7 (Dps), 8 (UspA), 9 (FtnA), 10 (Tnp), 11 (TrxA), 12 (CodY), 13 (RpoA), 14 (CysS), 15 (TrxB), 16 (Ldh1), 17 (PfkA), 18 (PdhB), 19 (CcpA), 20 (Trps), 21 (Pgl), 22 (PepP), 23 (Int), 24 (Hmp), 25 (TelA), 26 (Tuf), 27 (Eno), 28 (Gnd), 29 (Mqo2), 30 (PflB), and 31 (RpoB). **f** Silver contents in identified $Ag^+$-binding proteins. The silver contents were calculated by integrating the peaks of corresponding $Ag^+$-binding proteins. **g** GE-ICP-MS profiles of $Ag^+$-RpoA, $Ag^+$-Pgl and $Ag^+$-Gnd. **h** MALDI-TOF mass spectra of RpoA and Pgl without and with pre-incubation with $Ag^+$. **i** Gene Ontology (GO) analysis of biological processes enriched by $Ag^+$-binding proteins in *S. aureus* analyzed via STRING. Eleven biological processes were significantly enriched by GO analysis with $FDR < 0.05$. **j** Distribution of $Ag^+$ in different functional categories. The silver contents in identified proteins were summed according to the proteins involved in different pathways. The sum of silver contained in all identified $Ag^+$-binding proteins was set as 100%. (**a, b, c**) Mean value of three replicates is shown and error bars indicate ±SEM. (**e**, **g**, **h**) One representative of three replicates is presented.

number of functional classification systems. In total, 11 biological processes, 5 signaling pathways and 6 cellular components were significantly enriched (Fig. 1i and Supplementary Fig. 5, and Supplementary Table 6). In particular, these $Ag^+$-binding proteins are enriched in glycolytic process, ATP metabolic process, carbohydrate metabolic process, cellular nitrogen compound metabolic process, response to oxidative stress and translation. The enrichment analysis of signaling pathways showed that these identified proteins are involved in glycolysis/gluconeogenesis, pyruvate metabolism, microbial metabolism in diverse environments, carbon metabolism and ribosome (Supplementary Fig. 5 and Supplementary Table 7), generally consistent with the biological pathway enrichment analysis. The cellular component analysis revealed that 15 proteins are located in cytoplasm, 7 proteins are involved in forming macromolecular complex, and 5 proteins are ribosomal proteins with 2 of them being small ribosomal subunit (Supplementary Fig. 5 and Supplementary Table 8). The protein–protein interactions (PPI) analysis of identified $Ag^+$-binding proteins by STRING[44] demonstrated that these proteins form a highly connected network (Supplementary Fig. 6) with 36 nodes, 76 edges, and an average node degree of 4.22 ($P = 4.49E-0.5$), indicating $Ag^+$ exerts a global impact on the cell physiology of *S. aureus*.

We further calculated the $Ag^+$ contents contained in different proteins and biological processes by integrating the areas of corresponding $Ag^+$ peaks (Fig. 1j). The highest $Ag^+$ content was found in the biological process of response to oxidative stress (38.1%), followed by energy metabolism (20.8%). Both the functional enrichment analysis and $Ag^+$ distribution showed that $Ag^+$ exerts its antimicrobial effect against *S. aureus* mainly by disturbing oxidative stress homeostasis and energy metabolism.

**$Ag^+$ primarily targets glycolysis in *S. aureus* and induces ROS elevation at late stage**. We next evaluate how $Ag^+$ functionally disrupt the identified proteins in various pathways, among which glycolysis is the most significantly enriched biological pathway based on the bioinformatics analysis with four proteins are identified as $Ag^+$-binding proteins, i.e., ATP-dependent 6-phosphofructokinase (PfkA), enolase (Eno), pyruvate dehydrogenase E1 component subunit beta (PdhB), and lactate dehydrogenase (Ldh) (Fig. 2a). To examine the primary targets of $Ag^+$ among these proteins, we measured the activities of these enzymes in *S. aureus* cells post treatment of $Ag^+$ at different time points. The activities of Pfk and Eno were inhibited by over 50% from 5 min up to 1 h (Fig. 2b), suggesting they are potential primary targets of $Ag^+$. A similar time-dependent $Ag^+$-mediated inhibition of Pdh was noted with a slightly lesser extent. For Ldh, the activity was elevated to nearly 1.5-folds upon $Ag^+$ exposure for 5 min to 30 min but declined afterwards (Fig. 2b), with an overall decrease of the enzyme activities. The initial elevation of Ldh activity might be ascribed to the $Ag^+$-mediated activation of anaerobic respiration[45]. To exclude the indirect effect from different expression levels of the proteins under various conditions, we performed the real-time quantitative polymerase chain reaction (qRT-PCR) analysis to measure gene expression levels of the identified enzymes, which is generally in positive correlation with the abundance of the corresponding proteins. As shown in Supplementary Fig. 7, the genes coding for enzymes in glycolysis, e.g., *pfkA*, *eno*, and *pdhB*, were slightly upregulated upon treatment with $Ag^+$. More significant upregulation of *ldh* is noted, and the upregulation remained up to 1 h. Thus, it is unlikely that the $Ag^+$-mediated inhibition of these enzymes is induced by the downregulation of these proteins. The inhibition of these enzymes by $Ag^+$ was further validated by the abolishment of their activities after addition of 200 μM $AgNO_3$ to the cell lysate of *S. aureus* (Supplementary Fig. 8).

Given that $Ag^+$ disrupts glycolysis and binds to subunit alpha of ATP synthase (AtpA), the enzyme that produces the “energy currency” ATP for cells, we then determined the ATP concentration in *S. aureus* treated with different amounts of $AgNO_3$. Generally, a rapid dose-dependent $Ag^+$-induced depletion of ATP was noted (Fig. 2c). Specifically, addition of 20 μg/mL $AgNO_3$ to the cells led to the ATP abundance decreased by over 50% after 5 min, and nearly 70% after 60 min (Fig. 2c). These results collectively evidenced that $Ag^+$ primarily targets key enzymes in energy metabolism, particularly glycolysis in *S. aureus*.

Oxidative stress homeostasis is another pathway significantly enriched in GO enrichment analysis of $Ag^+$-binding proteins and with the highest silver content in the calculated silver distribution (*vide supra*). Several proteins, e.g., thioredoxin (TrxA), thioredoxin reductase (TrxB), alkyl hydroperoxide reductase AhpD (AhpD), and bacterioferritin comigratory protein (Bcp) that are utilized by *S. aureus* to respond to oxidative stress were identified as $Ag^+$-binding proteins (Fig. 2d). The alkyl hydroperoxide reductase activity was measured in *S. aureus* upon treatment of 20 μg/mL $AgNO_3$ for 1 h, and the enzyme activity was declined to 30% compared to untreated group (Supplementary Fig. 7), demonstrating that the binding of $Ag^+$ depletes the activity of

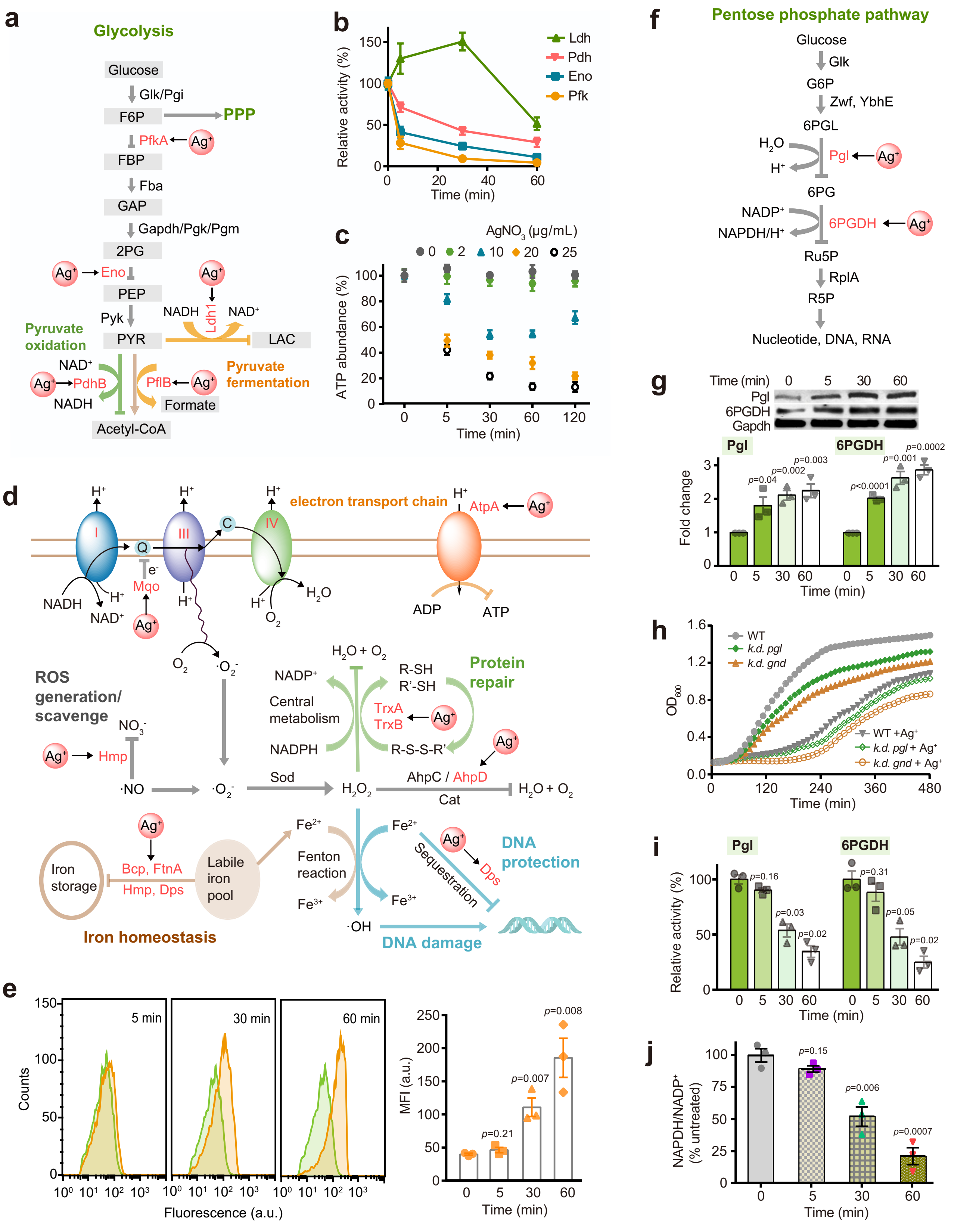


alkyl hydroperoxidase in *S. aureus* despite the significant upregulation of the genes coding for antioxidant enzymes in response to $Ag^+$ exposure (Supplementary Fig. 7). Together with previous reports that $Ag^+$ abolished the function of TrxA and TrxB both in vitro and in vivo[42,46], we unveiled a rather complete $Ag^+$-mediated damage on the oxidative stress defense system in *S. aureus*, which could result in accumulation of reactive oxygen species (ROS) and lead to the bacterial death. We therefore further measured the ROS levels in *S. aureus* by flow cytometry using the fluorescence probe 2′,7′-dichlorodihydrofluorescein diacetate (CM-$H_2$DCFDA)[47] (Fig. 2e and Supplementary Fig. 9). The calculated median fluorescence intensity (MFI) showed a

**Fig. 2 $Ag^+$ targets glycolysis, ROS defense system and oxidative pentose phosphate pathway of *S. aureus*.** **a** Diagram showing the identified $Ag^+$-binding proteins in glycolytic pathway and how $Ag^+$ disrupts glycolysis in *S. aureus*. G6P, glucose 6-phosphatase; F6P, fructose 6-phosphate; FBP, fructose 1,6-bisphosphate; GAP, glyceraldehyde 3-phosphate; 2PG, 2-phosphoglycerate; PEP, phosphoenolpyruvate; PYR, pyruvate; LAC, lactate. **b** Relative activities of glycolytic enzymes in *S. aureus* after treatment with 20 μg/mL $AgNO_3$ for different time points ($n = 3$). **c** Relative ATP abundance of *S. aureus* after treatment with 0, 2, 10, 20, and 25 μg/mL $AgNO_3$ ($n = 3$). The ATP contents in untreated *S. aureus* at corresponding time points were set as 100%. **d** Diagram showing how $Ag^+$ disrupts electron transport chain and oxidative stress defense system in *S. aureus*. **e** Measurement of ROS levels with flow cytometry ($n = 3$). Left: CM-$H_2$DCFDA fluorescence histogram of *S. aureus* with (yellow) or without (green) treatment of $Ag^+$ ($n = 3$). Right: mean fluorescence intensity. **f** Diagram showing how $Ag^+$ disrupts pentose phosphate pathway of *S. aureus*. G6P, glucose-6-phosphate; 6PGL, 6-phosphoglucono-δ-lactone; 6PG, 6-phosphogluconate; Ru5P, ribulose 5-phosphate; R5P, ribose 5-phosphate. **g** Expression level of Pgl and 6PGDH examined by western blotting ($n = 3$). Bar chart shows quantification of protein levels normalized to Gapdh and untreated control group. **h** Growth of WT *S. aureus* and mutants of *pgl* and *gnd* gene knock-down strains with or without treatment of 20 μg/mL $AgNO_3$ ($n = 3$). **i** Relative activities of Pgl and 6PGDH in *S. aureus* after treatment with 20 μg/mL $AgNO_3$ for different time points ($n = 3$). **j** NADPH/$NADP^+$ ratio in *S. aureus* exposed to 20 μg/mL $AgNO_3$ for various time points ($n = 3$). Mean value of three replicates is shown and error bars indicate ±SEM (**b**, **c**, **e**, **g**, **h**, **i**, **j**). Two-tailed *t*-test was used for all comparisons between two groups.

time- and dose-dependent increase of ROS in *S. aureus* upon $Ag^+$ exposure. For *S. aureus* after treatment of 20 μg/mL $AgNO_3$, no obvious increase of ROS (MFI = 45.9 a.u. vs control = 39.4 a.u.) was observed at 5 min, but a significant elevation of ROS (MFI = 110.3 a.u.) was noted after 30 min and became more significant after 60 min (MFI = 185.0 a.u.) (Fig. 2e), demonstrating that $Ag^+$ treatment induces ROS production at a later stage.

**$Ag^+$ targets oxidative pentose phosphate pathway**. It has been reported that glycolytic inhibition and elevation of ROS could promote the flux into the oxidative branch of the pentose phosphate pathway (oxPPP), which produces ribose 5-phosphate (R5P, a precursor for nucleotide synthesis) and nicotinamide adenine dinucleotide phosphate (NADPH)[48–50]. NADPH plays a critical role in maintaining cellular redox homeostasis since it provides the reducing power which recycles oxidized glutathione and fuels the protein-based antioxidant systems. Two key enzymes, i.e., 6-phosphogluconolactonase (Pgl) and 6-phosphogluconate dehydrogenase (Gnd or 6PGDH), from oxPPP were identified as $Ag^+$-binding proteins by LC-GE-ICP-MS (Fig. 2f). The expression levels of Pgl and 6PGDH were significantly elevated to 1.8 and 2.0-folds, respectively, in *S. aureus* exposed to $Ag^+$ for 5 min and the increase became more evident (2.1 and 2.6-folds for Pgl and 6PGDH, respectively) after 30 min (Fig. 2g). Such an elevation remained until 60 min with an overall Pgl being upregulated slightly less pronounced than 6PGDH (Fig. 2g). Consistently, the qRT-PCR results demonstrate a time-dependent upregulation of *gnd* and *pgl* genes in *S. aureus* exposed to $Ag^+$ (Supplementary Fig. 7), indicating that *S. aureus* cells activate oxPPP to response $Ag^+$ stress. We then compared the growth of the wild-type (WT) *S. aureus* and *pgl* or *gnd* gene knockdown strains (k.d. *pgl* and k.d. *gnd*) (Supplementary Fig. 10). The knockdown of either *pgl* or *gnd* gene led to slower cell growth compared to that of WT *S. aureus* (Fig. 2h), indicating the importance of these two genes for *S. aureus* survival. The addition $Ag^+$ to both *pgl* and *gnd* gene knockdown strains resulted in longer lag phases ($\lambda = 2.72$ and 3.71 h, respectively) than that of the WT strain ($\lambda = 2.35$ h), but no significant alteration of the maximum specific growth rate $\mu_{max}$ (Supplementary Fig. 11), demonstrating that these two enzymes are critical for *S. aureus* to alleviate $Ag^+$ toxicity (Fig. 2h).

However, in contrast to the elevated abundance of Pgl and 6PGDH, the activities of Pgl and 6PGDH decreased to 34.7% and 24.6%, respectively in *S. aureus* after treatment with $Ag^+$ for 1 h (Fig. 2i), suggesting the enzymes in oxPPP are severely damaged by $Ag^+$ ultimately. Thus, despite the activation of oxPPP in *S. aureus* in response to $Ag^+$ exposure, such a defense is largely futile, which is further supported by $Ag^+$-mediated inhibition of Pgl and 6PGDH in vitro (Supplementary Fig. 8). Since oxPPP is the major NAPDH-generating pathway in *S. aureus*, we therefore further measured the NADPH/$NADP^+$ ratio in *S. aureus* exposed to $Ag^+$. A sharp decrease in NADPH/$NADP^+$ ratio to 52.1% (versus untreated as 100%) was observed in 30 min and this became more evident at 60 min (21.0%) (Fig. 2j). Given that NADPH is a crucial antioxidant that quenches ROS to maintain the cellular redox homeostasis, the $Ag^+$-mediated deficiency of NADPH through disrupting oxPPP in *S. aureus* could result in hyperaccumulation of ROS and ultimate cell death.

**Mechanistic insights into $Ag^+$ inhibition of 6-phosphogluconate dehydrogenase**. As illustrated above, oxPPP was identified as a vital pathway targeted by $Ag^+$. As the third enzyme in oxPPP, 6PGDH converts 6-phosphogluconate (6PG) to ribulose 5-phosphate (Ru5P) with the concomitant reduction of $NADP^+$ to NADPH. Loss of 6PGDH activity is toxic to cells due to the accumulation of 6PG and deficiency of NADPH[51]. We then selected 6PGDH as a showcase to investigate how $Ag^+$ binds and functionally perturbs its protein targets.

Binding of $Ag^+$ to 6PGDH (monomer) was first examined by MALDI-TOF MS. The peak at *m/z* of 51842.3 was assigned to 6PGDH monomer, while the intense peak at *m/z* of 52379.6 was assignable to 6PGDH with five $Ag^+$ ions bound (*calcd m/z* of 52381.6) (Fig. 3a). No more $Ag^+$ binding was observed when further increasing the molar ratio of [$Ag^+$]/[6PGDH] to 12 (Fig. 3a). We then examined the inhibition of 6PGDH activity by $Ag^+$. A dose-dependent inhibition of $Ag^+$ on the activity of 6PGDH was observed and over 90% of the activity was inhibited by addition of 5 eq. $Ag^+$ (Fig. 3b). We further measured the effects of $Ag^+$ at different concentrations on the enzyme kinetics of 6PGDH. The apparent $V_{max}$ decreased from 1.1 to 0.15 μmol/min/mg when the molar ratio of $Ag^+$/6PGDH increased from 0 to 4; whereas the $K_m$ values of $Ag^+$-inactivated 6PGDH for 6PG were similar to that of the native enzyme (ca. 0.14 mM) (Fig. 3c), indicating $Ag^+$ exerts the inhibitory effect via a non-competitive inhibition with the rate determining step being affected but not the substrate binding.

We next investigate $Ag^+$ inhibition on 6PGDH at the atomic level by X-ray crystallography. Attempts on direct crystallization of Ag-bound 6PGDH were not successful. Thus, we crystallized apo-6PGDH, which stands for the first crystal structure of 6PGDH from *S. aureus*, and then soaked the crystals into cryoprotectants containing 0.5 mM $Ag^+$ for 2 h, allowing the protein to bind $Ag^+$ in its packed form. We used the structure of 6PGDH from *Geobacillus stearothermophilus* (PDB ID: 2W8Z) as the initial model for molecular replacement to solve the apo-form *Sa*6PGDH structure (PDB ID: 7CB0). The 7CB0 was subsequently used as the model for molecular replacement of substrate-bound (PDB ID: 7CB5) and Ag-bound 6PGDH structures (PDB

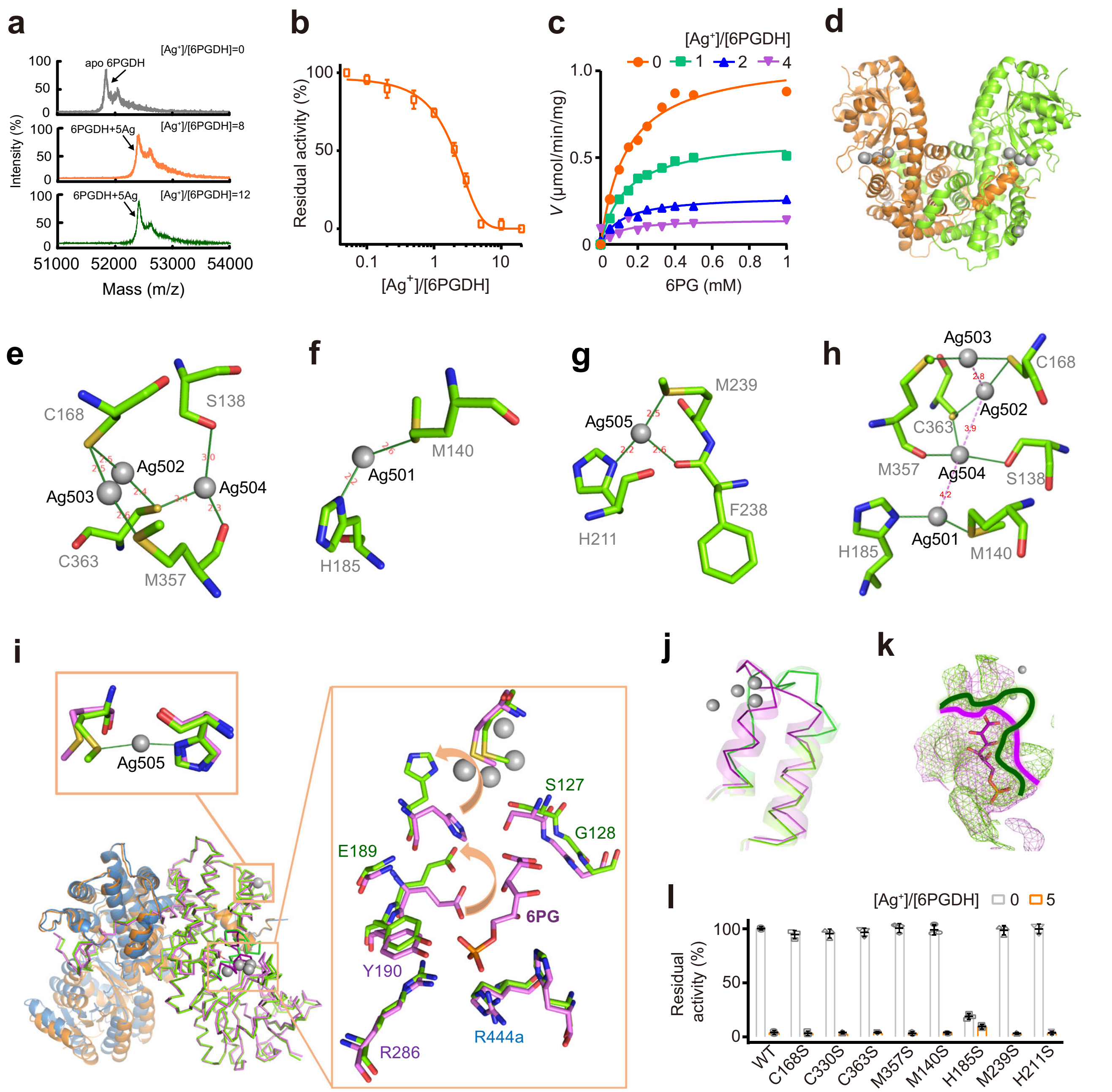

**Fig. 3 Molecular mechanism underlying $Ag^+$ inhibition of 6PGDH activity. a** MALDI-TOF mass spectra of apo-6PGDH and 6PGDH after incubation with 8 and 12 eq. of $Ag^+$. **b** Dose-dependent inhibition of 6PGDH by $Ag^+$ ($n = 3$). **c** 6PGDH activity in the presence of different amount of $Ag^+$. 6PGDH activity is defined as μmol/min/mg ($n = 3$). **d** Overall structure of Ag-bound 6PGDH. **e**–**g** Crystallographic analysis of the binding mode of $Ag^+$ at different coordination sites of 6PGDH. **h** Silver cluster composed by four adjacent Ag ions. **i** Superimposition of Ag-bound 6PGDH (green) to substrate-bound 6PGDH (violet). **j** Binding of Ag501-504 to 6PGDH significantly changed the secondary structure of residues of 352–371. Ag-bound 6PGDH and substrate-bound 6PGDH are presented in green and violet cartoon, respectively. **k** The 6PG binding pocket is apparently morphed in Ag-bound 6PGDH. Ag-bound 6PGDH and substrate-bound 6PGDH are presented in green and violet mesh, respectively. **d**–**k** The binding sites of $Ag^+$ in 6PGDH with $Ag^+$ shown as gray spheres. **l** Normalized residual activity of WT 6PGDH and 6PGDH mutants in the absence and presence of 5 eq. of $Ag^+$ ($n = 3$). Mean value of three independent replicates is shown and error bars indicate ±SEM (**b**, **l**).

ID: 7CB6). WinCoot[52] and Phenix[53] were used for the model-building and refinement. The Ag-bound 6PGDH was resolved at 2.6 Å resolution (Fig. 3d and Supplementary Table 9) with four polypeptide chains exist in each asymmetric unit. Identification of Ag was made on the significant positive peaks (≥20 σ) in the m*F*o-D*F*c (difference Fourier) map.

In total, five $Ag^+$ ions were observed in each 6PGDH monomer, among which, four of them are close to each other and three of them coordinate to Cys residues, i.e., Cys168 and Cys363 (Fig. 3e–g and Supplementary Fig. 12). In detail, Ag502 and Ag503 are bridged by the Sγ of Cys168 and bind to the Sγ of Cys363 or the Sδ of Met357, respectively, with both in quasi-linear coordination geometry (S-Ag-S angle of 157.8° and 166.0°, respectively) (Fig. 3e and Supplementary Table 10). Argentophilic interaction between Ag503 and Ag502 was observed with the Ag···Ag distance of 2.8 Å (Supplementary Table 11)[54]. Ag502 and

Ag504 are bridged by the Sγ of Cys363 whist Ag504 is further bound by the Oγ of Ser138 and the main-chain oxygen of Met357. Ag501 is bound to the Sδ of Met140 and Nδ1 of His185 in pseudo-linear coordination geometry (S-Ag-N angle of 131.5°) (Fig. 3f and Supplementary Table 10). Intriguingly, a silver cluster composed of four Ag ions, i.e., Ag501, Ag502, Ag503 and Ag504, and with Ag···Ag distance ranging from 2.8 to 4.2 Å is observed (Fig. 3h). At the site of Ag505, the Ag is coordinated with the Nδ1 of His211, Sδ of Met239, and main-chain oxygen of Phe238 (Fig. 3g), which is neither close to the other four Ag ions nor induces significant conformational change. In contrast, the binding of Ag501-Ag504 has significantly changed the secondary structure of residues of 352-371 when superimposing the Ag-bound 6PGDH to the 6PG-bound 6PGDH (Fig. 3i–k). Besides, two residues, i.e., His185 and Glu189, which are pivotal in stabilizing the substrate 6PG via hydrogen bonding, flips over for $Ag^+$ binding (Fig. 3i). The 6PG binding pocket is apparently morphed (Fig. 3k) due to the coordination of Ag ions to the surrounding residues despite these Ag ions are not located in the 6PG binding pocket directly.

To further depict which residue among those coordinated to Ag ions is critical for $Ag^+$-mediated inhibition of 6PGDH activity, we generated a series of 6PGDH mutants, including $6PGDH^{C168S}$, $6PGDH^{C330S}$, $6PGDH^{C363S}$, $6PGDH^{M357S}$, $6PGDH^{M140S}$, $6PGDH^{H185S}$, $6PGDH^{M239S}$, and $6PGDH^{H211S}$, and examined these enzyme activities without and with treatment of $Ag^+$. As shown in Fig. 3l, individual site-directed mutation of Cys168, Cys330, Cys363, Met357, Met140, Met239, and His211 to Ser led to no significant alteration of the 6PGDH activity. In contrast, the mutation of His185 to Ser resulted in a loss of ca. 80% of 6PGDH activity under identical conditions, implicating the essential role of this residue for the 6PGDH catalytic activity. Similar to the WT enzyme, addition of 5 eq. of $Ag^+$ abolished the activity of $6PGDH^{C168S}$, $6PGDH^{C330S}$, $6PGDH^{C363S}$, $6PGDH^{M357S}$, $6PGDH^{M140S}$, $6PGDH^{M239S}$, and $6PGDH^{H211S}$ (Fig. 3l), while for $6PGDH^{H185S}$, much less decrease in the enzyme activity (~50%) was observed, suggesting that His185 is the key residue for $Ag^+$-mediated inhibition of the enzyme activity.

Indeed, previous reports showed that the triad of Ser127/His185/Asn186 plays a vital role in placing the nicotinamide ring of the cofactor NADPH to interact with the substrate 6PG[55,56]. Thus, the binding of $Ag^+$ to His185 together with the morphing of the 6PG binding pocket could result in a loss of its interaction with the substrate and cofactor NADPH, thereby conferring the inhibition of 6PGDH activity, consistent with the non-competitive inhibition mode of $Ag^+$ against 6PGDH.

**Multi-target mode of action of silver against *S. aureus* suppresses its antibiotic selection effect**. Our combined data from metalloproteomics, systematic biochemical characterization, and structural biology demonstrated a multi-target mode of action of $Ag^+$, i.e., $Ag^+$ affects multiple pathways through functional disruption of protein targets against *S. aureus* (Fig. 4a), which is distinct from the conventional antibiotics with specific targets. We therefore hypothesize that silver could overcome resistance of *S. aureus* since the probability of simultaneous mutation on multiple targets is much less than on a single target. To validate the hypothesis, we compared the mutation frequency of *S. aureus Newman* under stress of $Ag^+$/AgNP and conventional antibiotics. The *S. aureus Newman* cells ($10^6$–$10^{10}$) were inoculated on nutrient plates with supplementation of $Ag^+$, AgNP (PVP-coated, with a diameter of 10 nm, Supplementary Fig. 13) and various antibiotics, including ampicillin (a $\beta$-lactam antibiotic), kanamycin (an aminoglycoside antibiotic), ciprofloxacin (a quinolone antibiotic), and tetracycline (a polypeptide antibiotic) at a concentration of 2× MICs. As shown in Fig. 4b, the mutation frequencies of resistance colonies evolved on agar plates for commonly used antibiotics ranging from $2.9 \times 10^{-8}$ to $6.3 \times 10^{-6}$. In contrast, no resistant colony was observed on agar plates with the supplementation of $Ag^+$ or AgNP (the mutation colony was then considered as less than one), and the mutation frequencies were calculated to be less than $1.0 \times 10^{-10}$ (Fig. 4b). We next examined whether *S. aureus* exhibited sustainable susceptibility to silver by serial passage through continuous exposing the cells to $Ag^+$ or AgNP. The continuous exposure of *S. aureus* to either $Ag^+$ or AgNP for 16 passages (*S. aureus*-P16) failed to select resistant mutants with reduced susceptibility as judged by no increase of the MIC values (Fig. 4c).

Different response of *S. aureus* to $Ag^+$/AgNP and commonly used antibiotics arouse our interest in further exploring the mechanism of action of silver-based antimicrobial agents towards inhibition of this pathogen. We further examined the $Ag^+$-binding protein profiles in *S. aureus*-P16 by LC-GE-ICP-MS. As shown in Fig. 4d, the $Ag^+$-proteome profile in *S. aureus*-P16 is similar to that of WT *S. aureus* (*S. aureus*-P0), with 29 out of 31 $^{107}Ag$ peaks being observed at the identical MW and pI despite the variation on the intensity of peaks (Fig. 4d, e). These results demonstrate that $Ag^+$ exerts the bactericidal activity against *S. aureus*-P16 through targeting same proteins and pathways as against *S. aureus*-P0, revealing the molecular mechanism underlying the sustainable susceptibility of *S. aureus* to silver.

**Sustainable susceptibility of *S. aureus* to silver endows it with potential to overcome antibiotic resistance**. The inability to select endogenous $Ag^+$-resistant *S. aureus* highlights the potential advantages of $Ag^+$/AgNP over conventional antibiotics for the treatment of staphylococcal infections. Since silver targets multiple pathways in *S. aureus*, we therefore reason that silver could potentiate the antibacterial activity of conventional antibiotics that share common mechanisms with silver. We next examined whether silver could resensitize MRSA to antibiotics. By using checkerboard microdilution assay, a typical synergistic effect between $Ag^+$/AgNP and antibiotics (ampicillin, kanamycin, tetracycline, and ciprofloxacin) against *S. aureus Newman* was observed with FIC index (FICIs) ranging from 0.1875 to 0.375 (Fig. 5a, b and Supplementary Fig. 14 a–c). The potent synergy was also demonstrated by time–kill curves that $Ag^+$/AgNP and ampicillin combinedly showed augmented bactericidal activity as reflected by 6–8-log decrease in the viability of *S. aureus Newman* compared to that of the control or single therapy for 24 h (Fig. 5c and Supplementary Fig. 14d).

The unique property of targeting multiple pathways by silver suggests its potential for combating antibiotic resistance. We then further measured the mutation prevention concentration (MPC) of antibiotic (using ampicillin as an example) in the absence or presence of different concentrations of $Ag^+$. We found that ampicillin alone, even at 8-fold MIC (MPC = 16-folds of MIC), was unable to kill high-level resistant mutants. In contrast, the number of mutant colonies declined significantly with the increase of $Ag^+$ concentration (Fig. 5d). The ampicillin MPC decreased to 0.5-fold MIC against *S. aureus Newman* when 8 μg/mL (75 μM) $Ag^+$ was used (Fig. 5e). Considering the rapid evolution of *S. aureus* into antibiotic-resistant variants, we further evaluated the effect of silver on the evolution of antibiotics by in vitro serial passage assay[57]. As shown in Fig. 5f, the resistance level to ampicillin was significantly elevated over 16 serial passages as the MIC increased dramatically by 16-folds. In contrast, such a phenomenon was not observed in the presence of

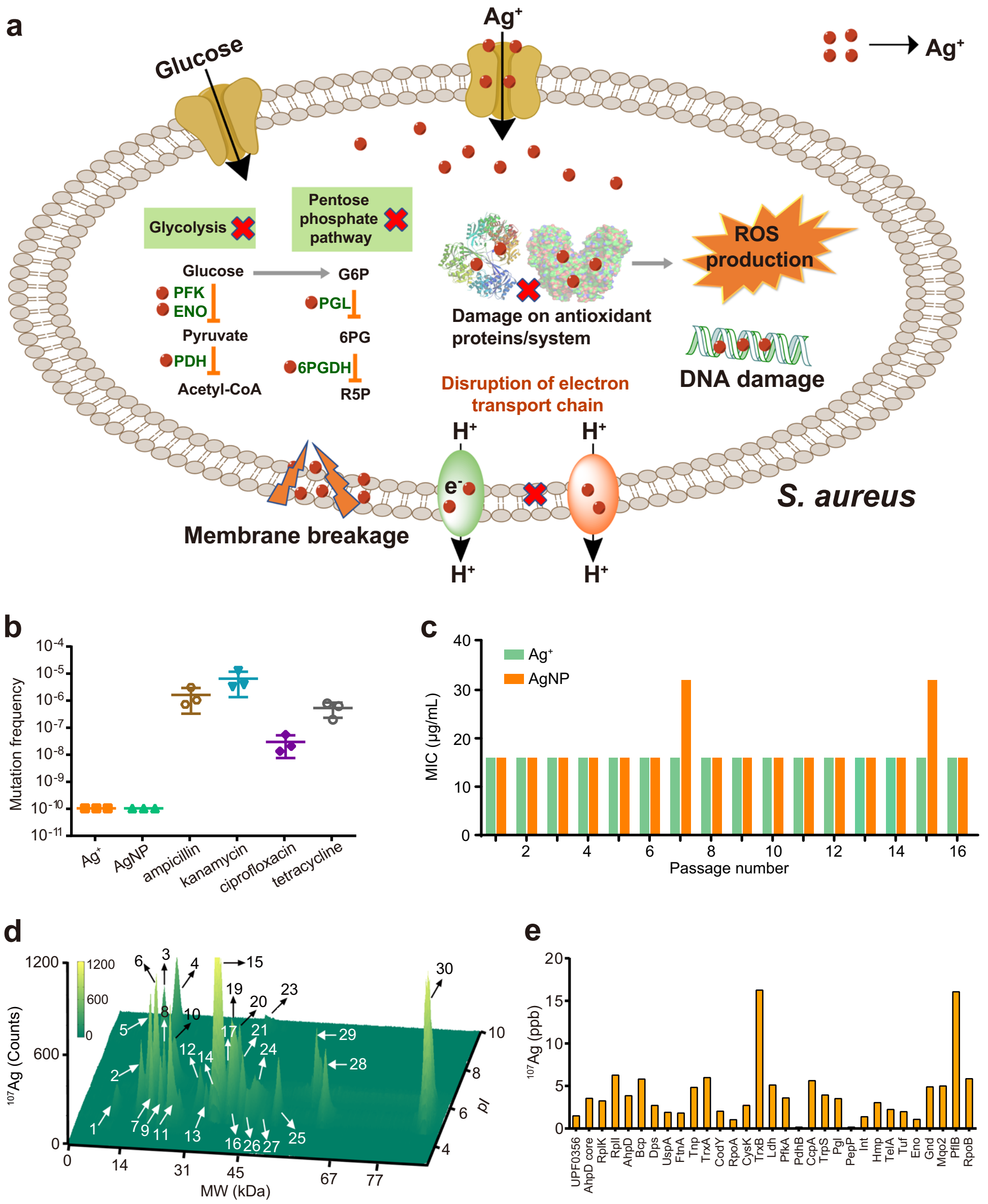


**Fig. 4 Multi-target mode of action of silver suppresses its antibiotic selection effect against *S. aureus*.** **a** Diagram showing that $Ag^+$ kills *S. aureus* by targeting multiple essential pathways. **b** Mutation frequencies of *S. aureus Newman* after treatment of $Ag^+$, AgNP, and different antibiotics at the concentration of 2×MIC ($n = 3$). **c** MIC of $Ag^+$ and AgNP against *S. aureus Newman* during passage for 16 times ($n = 3$). **d** LC-GE-ICP-MS profiles of $Ag^+$-binding proteins from *S. aureus*-P16. **e** Relative content of Ag in $Ag^+$-binding proteins resolved from *S. aureus*-P16. All experiments were performed in three biological replicates. Mean value of three replicates is shown and error bars indicate ±SEM (**b**).

$Ag^+$ (Fig. 5f), indicating that the combination therapy significantly suppressed the evolution of high-level resistant bacterial strains. The development of high-level ampicillin-resistant bacterial strains was also slowed down by AgNP as judged by only slight increase of MIC value to 4-folds (Supplementary Fig. 14e). Similarly, synergy between $Ag^+$/AgNP and ampicillin was noted against MRSA 33591 since the supplementation of 8 μg/mL $Ag^+$ or AgNP could decrease the MIC of ampicillin by 16-folds (Fig. 5g–i), suggesting that $Ag^+$/AgNP is able to resensitize MRSA to antibiotics. Collectively, we have demonstrated that the combination of antibiotics with $Ag^+$/AgNP has potential therapeutic applications in combating antibiotic-resistant *S. aureus*.

## Discussion

The steady increase in staphylococcal infections accompanied by rapid emergence of antibiotic-resistant *S. aureus* stimulates our demand to expand the antibacterial arsenal[58–60]. Reuse of metal-based antimicrobials to combat the current crisis of MRSA is a promising alternative strategy. With broad-spectrum antimicrobial activities, $Ag^+$ and AgNPs are widely used as an antimicrobial agent, whereas their internal usage is largely restricted owing to their potential toxicity to humans. The maximization of their therapeutic efficacy and minimization of the side effects depends heavily on the understanding its mode of action. Although enormous efforts have been devoted towards unveiling the molecular mechanism of the antibacterial activity of silver[9,17], proteome-wide identification of silver-binding targets have not been achieved till now owing to technical challenges and complexity of the metal–protein interactions. Previous quantitative proteomic studies allowed the profiling of up- and down-regulated proteins induced by $Ag^+$/AgNP exposure in *E. coli*[61,62], however, those regulated proteins might not serve as direct targets of silver. Unlike Gram-negative bacteria that can

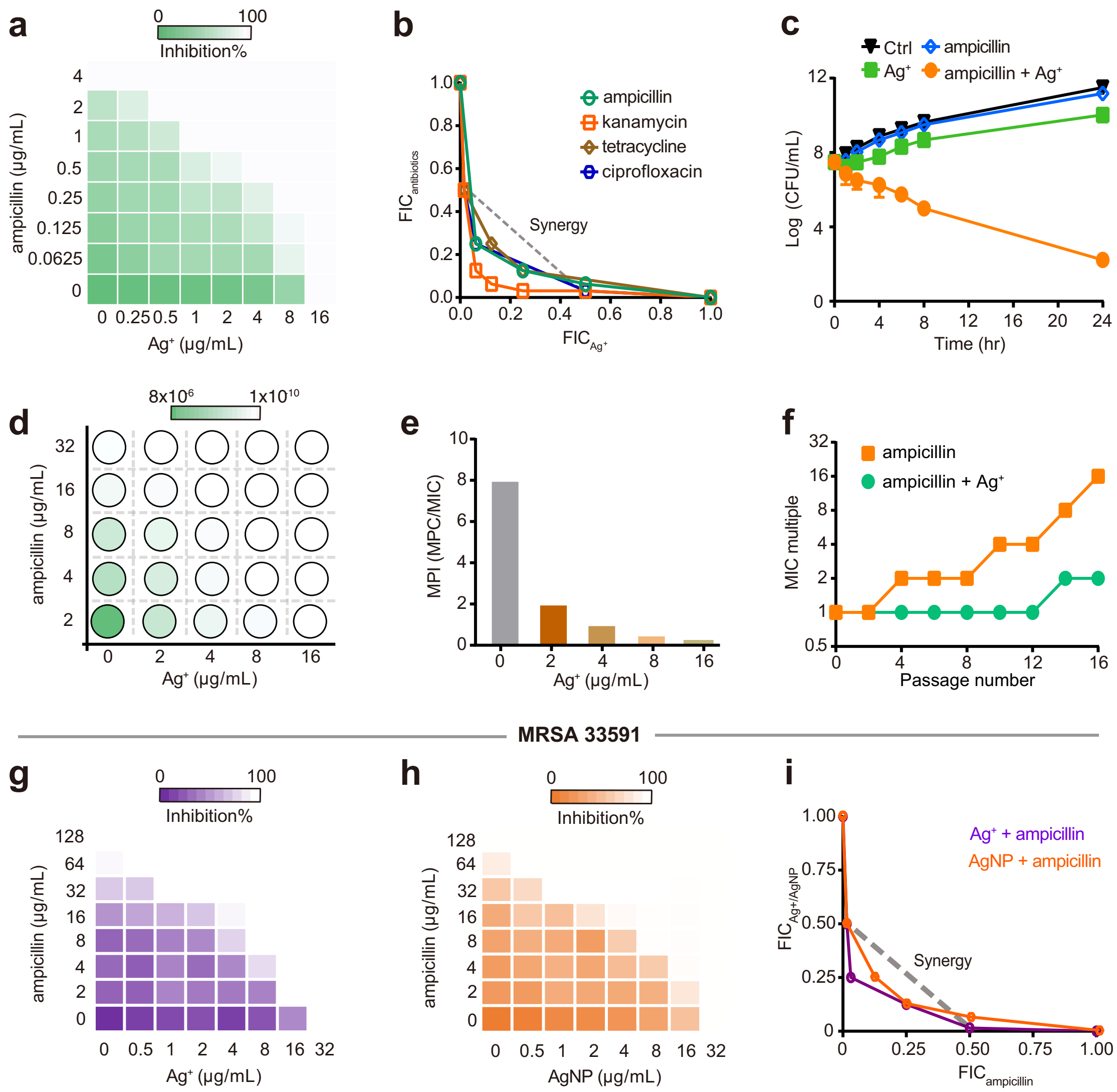


**Fig. 5 Sustainable susceptibility of *S. aureus* to silver endows it with potency to overcome antibiotic resistance.** **a** Representative heat plot of microdilution checkerboard assay for the combination of antibiotics and $Ag^+$ against *S. aureus Newman* ($n = 3$). **b** Isobolograms of the combination of ampicillin and $Ag^+$ against *S. aureus Newman* ($n = 3$). **c** Time-killing curves for ampicillin and $Ag^+$ mono- and combined-therapies against *S. aureus Newman* during 24 h incubation ($n = 3$). **d** Heat plot visualizing the mutation frequency of *S. aureus Newman* exposed to ampicillin in the presence of increasing concentration of $Ag^+$. **e** Bar chart showing MPI values of ampicillin in the presence of increasing concentration of $Ag^+$ against *S. aureus Newman* ($n = 3$). **f** Resistance acquisition curves during serial passage with the subinhibitory concentration of ampicillin or combination of ampicillin and $Ag^+$ against *S. aureus Newman* ($n = 3$). MIC test was performed every two passages. **g**–**i** $Ag^+$/AgNP synergize with antibiotics to kill MRSA 33591. **g** Representative heat plot of microdilution checkerboard assay for the combination of ampicillin and $Ag^+$ against MRSA 33591 ($n = 3$). **h** Representative heat plot of microdilution checkerboard assay for the combination of ampicillin and AgNP against MRSA 33591 ($n = 3$). **i** Isobologram of the combination of ampicillin and $Ag^+$/AgNP against MRSA 33591. All experiments were performed in three biological replicates. Mean value of three replicates is shown and error bars indicate ±SEM (**b**).

generate silver resistance through silver or copper efflux pump, Gram-positive bacteria exhibit no silver resistance[63]. Up to now, no authentic silver-binding proteins in Gram-positive *S. aureus* have been mined at a proteome-wide scale, which hinders systemic study on the biological pathways interrupted by silver.

Herein, using the customized approach of LC-GE-ICP-MS, we successfully separated and identified 38 authentic $Ag^+$-binding proteins ($Ag^+$-proteome) in *S. aureus* at the whole-cell scale. In combination with bioinformatics analysis and systematic biochemical characterization, we demonstrate that $Ag^+$ exploits a shotgun action through targeting multiple proteins, thus interfering with multiple pathways, including glycolysis, oxPPP, and ROS stress defense system, to exert its bactericidal effect against *S. aureus*. Further kinetic studies enable the snapshot on the dynamic action of $Ag^+$ at the molecular level to be captured, i.e., $Ag^+$ primarily targets glycolysis via inhibiting multiple essential enzymes and induces the elevation of ROS through functional disruption of redox homeostasis system at the late stage, leading to an upregulation of oxPPP enzymes to alleviate $Ag^+$ stress. However, such a defense via activation of oxPPP is ultimately futile due to key oxPPP enzymes of Pgl and 6PGDH being inhibited by $Ag^+$.

Among the enzymes involved in these pathways, three enzymes from glycolysis (Pfk, Eno, and Pdh) and two enzymes from oxPPP (Pgl and 6PGDH) are newly identified protein targets of $Ag^+$ by our in-house developed LC-GE-ICP-MS, of which $Ag^+$-induced impairment of their functions are systematically examined in this study. Besides, since many ribosomal proteins

are also identified as $Ag^+$ binding proteins, we further measured the effect of $Ag^+$ on nascent protein synthesis of *S. aureus* via *O*-propargyl-puromycin (OPP) labeling and Click-iT chemistry[64]. As shown in Supplementary Fig. 15, the fluorescence signals from OPP which is incorporated into the newly synthesized peptides were decreased with the increase in $Ag^+$ concentration, demonstrating $Ag^+$ inhibits the nascent protein synthesis of *S. aureus* in a dose-dependent manner despite the upregulation of *rpoA* and *rpoB*, consistent with our findings by metalloproteomic approach. Together with the previously validated $Ag^+$ targets of CcpA[41], TrxA[42], TrxB[42], and Ldh[43], a rather dynamic and complete $Ag^+$ poisoning of the central carbon metabolism, ROS defense system, oxidative pentose phosphate pathway, and translation, in *S. aureus* is unraveled (Supplementary Fig. 16). It is noteworthy that we hereby provide direct evidence to define how $Ag^+$ exposure leads to increased ROS levels in *S. aureus*, i.e., $Ag^+$ functionally disrupts antioxidant enzymes and ultimate damages the oxidative repair system, which clarifies the conflicting results on ROS production by Ag redox chemistry.

Among these pathways, we uncover that oxPPP serves as a vital pathway targeted by $Ag^+$ in *S. aureus*. With the recombinant 6PGDH, we show that $Ag^+$ binds to 6PGDH and inhibits its activity in a non-competitive manner. More importantly, we resolved the first crystal structures of 6PGDH from *S. aureus* both in substrate-bound and Ag-bound forms. Those structures reveal that $Ag^+$ binds to 6PGDH at both catalytic and non-catalytic sites with dominant binding residues of Cys, His and Met in quasi-linear and trigonal geometry, which is generally consistent with our previous reports[36,40]. Together with the site-directed mutagenesis study, we unveil that $Ag^+$ abolishes the enzymatic activity of 6PGDH through targeting His185 in the active site and morphing its catalytic pocket.

Although it has been reported that silver could enhance the antibacterial activity of antibiotics[17,65], the molecular mechanism underlying the sustainable efficacy of silver against pathogens remain obscure. Our study resolves the long-standing question on the molecular targets and mode of action of silver against *S. aureus*. Such a unique mode of action of $Ag^+$ via targeting multiple pathways confers the inability to select $Ag^+$-resistant *S. aureus* and endows it with the sustainable efficacy against *S. aureus*. Based on the uncovered molecular mechanism, we further demonstrate that $Ag^+$/AgNP can potentiate the efficacy of a broad range of antibiotics, resensitize MRSA to antibiotics, and slow down the evolution of antibiotic resistance in *S. aureus*, highlighting a proof-of-principle that fundamental knowledge and molecular understanding of $Ag^+$/AgNP toxicity can be translated into antibacterial approaches in clinic. Therefore, combination of antibiotics with silver or other metal-based compounds or nanomaterials, e.g., colloidal bismuth subcitrate (CBS)[20] and auranofin[66], could serve as a promising strategy to suppress the selection effects of antibiotics, thus preventing occurrence of primary antibiotic resistance and extending lifespan of conventional antibiotics to relieve the current crisis of antibiotic resistance.

Nevertheless, given a large portion of essential enzymes are conserved among humans and pathogens, it is likely that silver ions and silver nanoparticles exhibit a poor selectivity to these enzymes in *S. aureus* over those in humans. Thus, despite of the enormous efforts being devoted, it remains to be a huge challenge to improve the biocompatibility and reduce the side effect of silver[23,67]. Finally, the interactions between silver and the rest identified $Ag^+$-binding proteins warrant further investigation, which may allow the identification of protein targets for antibiotic discovery.

## Methods

**Materials and experimental design**. *S. aureus* strain Newman is from our lab collection and MRSA 33591 is a gift from Dr. Wei Xia (Sun Yat-sen University). The FPLC system and Mono P 5/50 GL column were purchased from GE Healthcare. To rebuild the column gel electrophoresis separation system, the gel column (Φ = 2.5 mm) and solution transfer tubing (Φ = 0.25 mm PEEK tubing) were replaced from the Mini Prep Cell system (Bio-Rad). All the ICP-MS detections were performed on Agilent 7700X system. All standard proteins and the chemical reagents, unless otherwise specified, were purchased from Sigma-Aldrich. Milli-Q water (Milli-Q Ultrapure water systems, Millipore, USA) was used to prepare all solutions. Peptide mass fingerprinting was performed on an LTQ Orbitrap VelosTM mass spectrometer (Thermo Scientific).

**Cell culture and antibacterial activity test of $AgNO_3$ against *S. aureus***. *S. aureus Newman* stocks were inoculated to BHI agar plate and cultured for overnight at 37 °C. Single colonies from the agar plate were added to BHI broth and grown for overnight at 37 °C with agitation (200 rpm). The overnight cell cultures were then diluted (1:100) to BHI broth and grown for 2 h to the early-log phase ($OD_{600}$: 0.2–0.3). The minimum inhibitory concentration (MIC) of $AgNO_3$ against *S. aureus* were routinely determined by 2-fold serial dilution in BHI according to the reference[39]. The growth curves of *S. aureus* upon exposure to $AgNO_3$ with different concentrations were determined by measuring $OD_{600}$.

**Silver uptake measurement**. The early-log phase ($OD_{600}$ of 0.2–0.3) *S. aureus* cells after treatment of various concentrations of $AgNO_3$ for different time points were harvested by centrifugation (4500*g* and 4 °C for 10 min), and washed with cold PBS for 3 times. *S. aureus* cells with the number of 1 OD were collected and followed by digestion with 100 μL $HNO_3$ (69.0%, Fluka, 84385). The digested cell solutions were then diluted to a final volume of 1 mL with 1% $HNO_3$ and were further diluted when the signals exceed the range of standard curve. Multielement standard solution (90243, Sigma-Aldrich) was used to prepare standard curve of silver with the concentration of 0, 5, 10, 20, 50, and 100 ppb. Silver contents in *S. aureus* were calculated based on the standard curve and normalized to atom number in single cell.

**Protein extraction and silver content measurement**. *S. aureus* cells after treatment with 20 μg/mL $AgNO_3$ for 5, 30, 60, and 120 min were harvested by centrifugation (4500*g*, 4 °C, and 15 min). The pellets were then washed with chilled PBS for 3 times and then resuspended in PBS containing 1 mM TCEP and 0.5 mM PMSF. The cells were then lysed via sonication at the amplitude of 25% on the ice-water bath (5 s on, 10 s off, in total 20 min). The sonicated cells were centrifuged at 4500*g* for 10 min and the supernatant was further centrifuged at 100,000*g* for 10 min to separate the soluble (the supernatant) and membrane proteins (the pellet). The pellets were dissolved with PBS buffer containing 1% (w/v) SDS to get the membrane proteins[68], while the remaining pellet that cannot be dissolved by 1% SDS, which mainly composed by cell wall, is defined as the others.

Pierce™ BCA Protein Assay Kit (Thermo Scientific) and ICP-MS were used to measure protein concentrations and silver contents in soluble and membrane proteins, respectively. For the distribution of silver in different fractions, same amounts of *S. aureus* cells were used to measure the total silver content and the percentages were obtained by dividing the silver content in different fractions to total silver content.

**Silver-binding protein separation and identification**

*First dimensional separation with liquid chromatography*. *S. aureus* soluble proteins (0.5 mg) were used to explore $Ag^+$-binding proteins. The samples were changed into start buffer for 3 times. Conditions utilized were referred to standard procedures and further optimized. Polybuffers were diluted 16-fold with milli-Q water before adjustment of pH. The pH range of 3.5–6.5 was used instead of 4.0–7.0 as the pI of more than half of proteins are less than 6. For the pH range of 3.5–6.5, iminodiacetic acid was used to adjust pH and 0.025 M bis-Tris (pH 7.1) and polybuffer 74 (pH 4.0) were selected as starting buffer and eluent, respectively. As for the pH in the range of 6–9, acetic acid ($CH_3COOH$) was chosen to adjust pH of start buffer (0.075 M Tris) and eluent (polybuffer 96) to 10.0 and 6.5, respectively. The fractions of FPLC were collected and condensed to same volume for each 0.5 pH and subsequently applied to GE-ICP-MS for further separation and detection.

*Second dimensional separation and detection*. The apparatus and operation procedures followed the previous work with modification[36–38]. The Tris/glycine/SDS buffer (0.025 M Tris, 0.192 M glycine, 0.1% SDS (w/v)) and 50 mM ammonium solution were used for running gel column and eluting separated proteins, respectively. The proteins were subjected to ICP-MS via a peristaltic pump equipped with 0.25 mm pump tubing. To reduce the effect of generated heat during the electrophoresis, a cooling system was applied to the gel column and the whole-gel electrophoresis chamber was kept in the cooling box during separation and detection.

In general, 10 μL of protein samples were mixed with 2 μL of loading buffer (SDS-PAGE: 0.5 M Tris buffer, pH = 6.8, 50% glycerol (v/v), 2% SDS (w/v)) and then loaded to the stacking gel. The concentration and length of freshly prepared column were optimized according to the classical slab SDS-PAGE gel system. All fractions from FPLC were subjected to the same separation conditions. In detail, the length of 15% and 12% gels were increased to 1.7 and 2.0 cm, respectively, and

the length of stacking 4% gel was doubled to 1.2 cm. A two-step voltage program was used for the separation with a consecutive procedure of 60 min at 200 V, and the rest at 600 V. No protein came out from the gel column during the first step, and the elemental detection by ICP-MS was started at the beginning of the last voltage step. For each GE-ICP-MS experiment, same amount of iodine-labeled standard proteins was selected as internal standards for calibration of MWs and intensities, avoiding the interference of metals from the standard proteins.

A T-connection was employed to split the solution after gel electrophoresis separation system, and additional pump tubing with the same diameter was used to transfer half of the split solutions to a collection tube. After reducing sample volume with ultrafiltration device (MW cut-off of 3 kDa), a one-dimensional gel electrophoresis was used to separate the collected fractions, which would be of benefit to concentrate the target proteins as well as become more compatible with mass spectrometry for subsequent protein identification than kept in solution. In case that multiple protein bands were obtained in a fraction, an additional metal determination procedure was applied to verify the associated metal in each band.

**Protein identification**. The proteins in the collected fractions after gel electrophoresis separation were identified through peptide mass fingerprinting.

*Extraction and trypsin-digestion*. The destained gel pieces were first rinsed with MilliQ water, 50% acetonitrile (ACN), then by 10 mM ammonium bicarbonate and dehydrated with 100% ACN. The gel plugs were rehydrated with 12.5 ng/μL trypsin (Promega) with volume that was sufficient to cover the gel pieces (e.g., ~50 μL) in 10 mM ammonium bicarbonate. After incubation for 16 h at 37 °C, the peptides were extracted twice using 5% formic acid (FA)/50% ACN and then extracted once with 100% ACN. After drying in a SpeedVac concentrator (Eppendorf), the peptides were resuspended in 0.1% FA, and purified using μC-18 Ziptips (Millipore). The desalted peptides were then mixed in a 1:1 ratio with 10 mg/mL α-Cyano-4-hydroxycinnamic acid matrix (Fluka) dissolved in 0.1% FA/50% ACN.

*Mass spectrometric analysis*. The apparatus and operation procedures followed our previous work[38]. The 4800 MALDI TOF/TOF Analyzer (ABSciex) equipped with a Nd:YAG laser operating at 355 nm for sample ionization was used for protein identification and characterization. The 4000 Series Explorer version 3.5.28193 software (ABSciex) was used to acquire all the mass spectra in the positive ion Reflector mode. The MS and MS/MS were calibrated using the peptide calibration standard (4700 Cal-Mix, ABSciex) that was spotted across 13 locations on the MALDI target plate. To keep the best resolution, characteristic spectra were attained by averaging 500 acquisitions in Reflector and MSMS mode with the smallest laser energy. Precursor ions with 1+ charge state were fragmented by post-source decay. Each sample was first analyzed with MALDI-TOF MS to create the PMF data in the scanning range of 900–4000 *m/z*. To make the full scan mass spectrum, the five most abundant peptides (precursors) were chosen for advance fragmentation (MALDI-TOF/TOF) analysis. Minimum S/N filter of 20 was used as monoisotopic precursor selection criteria for MSMS. The peak detection criteria for MSMS used were an S/N of 5 and a local noise window width of 250 (*m/z*) and a minimum full-width half-maximum (bins) of 2.9. The GPS Explorer algorithm version 3.6 (ABSciex) and in-house MASCOT search engine version 2.2 were used for combined PMF and MS/MS search against the non-redundant NCBInr database. The criteria for protein identification were based on the probability score of each search result and the significant matches had scores higher than the minimum threshold set by MASCOT ($P < 0.05$). All the results are presented in Supplementary Tables 15–52.

**Bioinformatics analysis**. The free-web STRING database (version 9.1, http://string-db.org/) was applied to analyze the statistically enriched Gene Ontology (GO) terms and PPI of identified proteins[44]. The GO enrichment analysis was carried out to identify statistically enriched GO categories in the $Ag^+$-binding proteins. The false discovery rates (FDR) were calculated with the number of *S. aureus* proteins as the background list. The enriched GO terms were selected with significant threshold of FDR < 0.05. The enriched biological process, KEGG pathways and cellular components with FDR less than 0.05 were presented.

For the PPI networks mapping, a dataset containing sequences of all the identified proteins were uploaded to STRING. The significantly enriched PPI networks were determined with the default parameter settings and the PPI spider employs statistical methodology with *P*-values being adjusted for multiple testing (Bonferroni correction). The statistical significance of the inferred model was calculated according to the distribution of the model size for a random protein list. To visualize the PPI networks among identified $Ag^+$-binding proteins[69], the Cytoscape software (version 3.4.0, http://www.cytoscape.org/) was used and all the nodes involved were labeled with corresponding gene name.

**Enzymatic activity measurement**. Commercially available colorimetric assay kits of enolase (Abcam), pyruvate dehydrogenase (Abcam), phosphofructokinase (Abcam), and lactate dehydrogenase (Abcam) were used to measure corresponding enzymatic activities. In brief, *S. aureus* cell cultures with or without treatment of $AgNO_3$ were harvested, washed, suspended, and sonicated in lysis buffers. The protein concentration of supernatant after centrifugation was measured by BCA assay. Enzyme activities were measured according to the standard procedures provided by the manufacture. The enzymatic activity was normalized to protein concentration. The activity of alkyl hydroperoxide reductase[70], Pgl[71], and 6PGDH[72] were performed according to previous reports.

**ATP concentration detection**. Cellular ATP concentrations of *S. aureus* cells exposed to $AgNO_3$ were measured with CellTiter-Glo Luminescent Cell Viability Assay (Promega) according to the manufacturer's instruction. In the luciferase reaction mono-oxygenation of luciferin is catalyzed by luciferase in the presence of $Mg^{2+}$, ATP and molecular oxygen. Generally, *S. aureus* cells were cultured in BHI medium and grown for 2 h to the early-log phase ($OD_{600} \sim 0.2$) and then treated with 0, 2, 10, 20, and 25 μg/mL $AgNO_3$. After treatment for certain time points, *S. aureus* cell were harvested, washed with chilled PBS for 3 times and diluted till $1 \times 10^7$ cells/mL based on OD estimation. 100 μL of CellTiter-Glo reagent were added to a 100 μL aliquot of the culture. The mixture was incubated at room temperature and in dark for 15 min and then the luminescence was measured on PerkinElmer 2030. A standard curve was simultaneously constructed using six concentrations of ATP in the range of 0.125–1 μM.

**ROS detection**. The general ROS indicator of 2′, 7′-dichlorodihydrofluorescein diacetate (CM-$H_2$DCFDA) (Thermo Scientific)[73] was utilized to examine the ROS induced by $AgNO_3$ in *S. aureus*. CM-$H_2$DCFDA is a chloromethyl derivative of $H_2$DCFDA with improved retention in live cells. CM-$H_2$DCFDA can diffuse into cells passively, where its acetate groups are cleaved by intracellular esterase and its thiol-reactive chloromethyl group reacts with intracellular glutathione and other thiols. Subsequent oxidation yields a fluorescent adduct ($\lambda_{ex}/\lambda_{em}$: 492-495/517-527 nm) that is trapped inside the cell. Briefly, *S. aureus* cells ($10^8$ cell/mL) after treatment with $AgNO_3$ for different time were incubated with CM-$H_2$DCFDA (final concentration of 10 μM) for 30 min at 37 °C in dark.

ROS levels in single *S. aureus* cells were detected by flow cytometry with BD FACS AriaIII flow cytometer (Becton Dickinson, USA). CM-$H_2$DCFDA was excited by 488 nm laser and detected at 535 nm. Forward and side scatter gates was established to exclude debris and cellular aggregates from analysis. Cell density were adjusted to around $1 \times 10^6$ cells/mL to avoid overly dense during analysis. 10,000 cells were analyzed per experimental condition. Data were processed with FlowJo (version 10.0). Debris and isolate cell population of interest were excluded during gating. Mean fluorescent intensity (MFI) were used to determine fold change between untreated and treated groups.

**Gene knockdown and growth curve measurement**. To knockdown the genes of *gnd* and *pgl*, the 5′-NGG-3′ sequence was identified in the double strand. The 23 nt immediately upstream of the 5′-NGG-3′ were then taken and added to the 3′ end of 5′-CTA-3′ to create sgRNA oligonucleotide 1. The reverse complementary sequence of the 23 nt was then taken and added to the 3′ end of 5′-AAC-3′ to create sgRNA oligonucleotide 2. The 11 nt upstream of NGG were BLAST searched against whole genome to confirm the specificity of the sgRNA. The two oligonucleotides with the concentration of 20 μM were subsequently annealed and cloned into SapI-digested plasmid pSD1[74]. The plasmids were introduced into *S. aureus* strain RN4220 and then transduced into Newman with phage Φ85. The sequences of the primers and sgRNA oligonucleotides used for plasmid construction are shown in Supplementary Table 14.

For the measurement of growth curve, single colonies of WT *S. aureus* Newman or its mutant strains were picked from BHI agar plate and cultured in BHI medium supplemented with 100 ng/mL tetracycline, which was added to induce gene knock down. The culture was diluted 10 times with fresh LB medium when $OD_{600}$ of the culture cells reached 0.5. The *S. aureus* cultures were then incubated at 37 °C in a 24-well plate with orbital shaking at 220 rpm. The *S. aureus* growth was monitored via $OD_{600}$ values at 10 min intervals. For the $Ag^+$-treatment, 20 μg/mL $AgNO_3$ was added.

**Western blotting analysis**. Western blotting analysis was performed according to the standard protocol. In detail, *S. aureus* cells after treatment with various conditions were harvested and lysed through cell lysis buffer. Cell debris was removed by centrifugation (16,000*g* for 20 min at 4 °C). The supernatant was collected, and the protein concentration was determined by BCA assay. Generally, 20 μg of proteins were separated by polyacrylamide gel electrophoreses (SDS-PAGE). After SDS-PAGE separation, proteins of interest were transferred to PVDF membranes using 90 V constant voltage for 1.5 h. Membranes were blocked with 5% (w/v) BSA in TBST buffer for 2 h at room temperature, and subsequently incubated with primary antibodies overnight at 4 °C with optimized dilution ratios. After washing with TBST buffer for 3 times, the membranes were further incubated with diluted secondary antibodies for 2 h at room temperature and incubated in chemiluminescent substrate working solution (Thermo) after washing for the detection of HRP on the membranes. The antibodies used were rabbit anti-Pgl (1:2000 dilution), anti-6PGDH (1:2000 dilution) and anti-Gapdh (Abcam, ab181602, 1:10,000 dilution). The anti-Pgl and anti-6PGDH antibodies were purified from the antibodies produced via injection Pgl or 6PGDH proteins into rabbits. Immunoreactive bands were visualized by the enhanced chemiluminescence detection system and the intensity of bands was quantified using a model GS-700 Imaging Densitometer (Bio-Rad). Protein band densities were analyzed using ImageJ software.

**Determination of NADP$^+$/NADPH ratios**. EnzyChrom™ NADP$^+$/NADPH Assay Kit (BioAssay Systems, USA) was used to measure NADP$^+$/NADPH ratios. In brief, *S. aureus* cells were cultured to $OD_{600}$ of 0.3 and then treated with 20 µg/mL $AgNO_3$ for various time points. *S. aureus* cell cultures with or without treatment of $AgNO_3$ were collected at different time points, washed, re-suspended with cold PBS, and adjusted to $OD_{600}$ of 3.0. For each measurement, aliquots of 0.5 mL *S. aureus* cells were sonicated and centrifuged. The collected supernatant after deprotonation by 3 kDa cut-off filters (Amicon) was used to measure NADP$^+$/NADPH ratios. The assay kit is based on a glucose dehydrogenase cycling reaction, in which the formed NADPH reduces formazan and the intensity of the reduced product color is proportionate to the NADP$^+$/NADPH concentration. The standard curve was obtained using $\beta$-NADP$^+$ (Sigma) and water was used as blank. The reactions were monitored at 565 nm for 30 min using a SpectraMax iD3 Multi-Mode Microplate Reader (Molecular Devices).

**RNA extraction and qRT-PCR**. Quantitative real-time polymerase chain reaction (qRT-PCR) was performed with Superscript III (Invitrogen) according to the manufacturer's protocol. The total RNAs of *S. aureus* were extracted using RNeasy Mini Kit (Qiagen) according to the protocol. The extracted RNA samples were subjected to DNase I treatment using the Turbo DNA Free Kit (Ambion) to completely remove contaminated genomic DNA. PCR was used to confirm the absence of genomic DNA contamination with the prepared RNA as a template. The quantity and integrity of RNA were determined by NanoDrop 2000 (Thermo Fisher Scientific) and verified by agarose gel electrophoresis.

Super Script$^{TM}$ II reverse transcriptase (Invitrogen) and random hexamer primers (Invitrogen) were used for reverse transcription. Real-time PCR reactions were performed on StepOnePlus$^{TM}$ Real-Time PCR system (ABI) with the SYBR Green qPCR Master Mix (Thermo Fisher Scientific). $\Delta\Delta C_T$ method was utilized to quantify the transcription level of target genes and the levels were normalized to the house-keeping gene *rrsA*. Each sample was assessed in three biological replicates and two technique replicates including non-template control. The primers used for qRT-PCR are listed in Supplementary Table 14.

**Nascent protein synthesis assay**. To quantify the effect of $Ag^+$ on nascent protein synthesis of *S. aureus*, OPP labeling method was used and followed by Click-iT chemistry detection of nascent peptides[64]. A commercial assay kit was used according to manufacturer's protocol (Click-iT™ Plus OPP Alexa Fluor™ 647 Protein Synthesis Assay Kit, Molecular Probes, C10458). In brief, *S. aureus* cells (at the early-log phase with $OD_{600}$ of 0.3) after treatment with 20 µM OPP and various concentrations of $AgNO_3$ for 1 h were harvested and washed with cold PBS. The cells were further proceeded to fixation by 3.7% formaldehyde, permeabilization by 0.5% Triton X-100, and incubation with the click reaction cocktail for 30 min. The fluorescence ($\lambda_{ex} = 590$ nm and $\lambda_{em} = 617$ nm) of *S. aureus* cells with cell number of 1 OD were measured on SpectraMax iD3 (Molecular Devices, USA).

**Strains, plasmids, and primers for protein expression**. Strains, plasmids, and primers used for protein overexpression are listed in Supplementary Tables 12 and 13. The *E. coli* XL1-Blue and BL21(DE3) strains harboring designed vectors were cultured in LB medium supplemented with appropriate concentration of ampicillin.

**DNA manipulation and plasmid construction**. Plasmid extraction kit (QIAprep Spin Miniprep kit, QIAGEN) was used to extract all the plasmids used as templates for PCR. All PCR primers were synthesized by BGI Company (Guangdong, China). Genes of *gnd*, *rpoA*, and *pgl* were amplified by PCR using *S. aureus* Newman chromosomal DNA as a template with the primers listed in Supplementary Table 13. These amplified genes contain AgeI and EcoRI restriction site at 5′- and 3′-end, respectively. The corresponding amplified products were digested with AgeI and EcoRI and ligated into pHisSUMO plasmid being digested with AgeI and EcoRI. The resulted plasmids pHisSUMO-*gnd*, pHisSUMO-*rpoA*, and pHisSUMO-*pgl* were extracted and transformed into BL21(DE3) cells for protein expression.

**Protein expression and purification**. The BL21(DE3) cells harboring pHisSUMO-*gnd*, pHisSUMO-*rpoA* and pHisSUMO-*pgl* plasmids were cultured overnight, and then diluted by 1:100 to fresh LB medium supplemented with 100 µg/mL ampicillin. Cells were grown (37 °C and 200 rpm) to $OD_{600}$ of 0.6 and then induced 6PGDH expression was induced by 20 µM $\beta$-D-thiogalactoside (IPTG) and the bacteria were further incubated at 18 °C for 20 h. For Pgl and RpoA, protein expression was induced by 500 µM IPTG and further cultured for 20 h at 25 °C. The bacteria were harvested by centrifugation (5000*g*, 4 °C for 20 min) and the cell pellets were resuspended with 50 mM Tris-$HNO_3$ buffer (150 mM $NaNO_3$, pH = 7.4) and lysed by sonication. The lysates were centrifuged (15,000*g*, 4 °C for 30 min). The supernatant was then collected and applied to a 5 mL HisTrap Q column (GE Healthcare). The proteins were eluted with 300 mM imidazole in the same buffer and the eluted proteins were further subjected to SUMO protease (50 NIH units) cleavage at 25 °C for 2 h. After removing the His-tag, the proteins were further purified by HiLoad 16/60 Superdex 200 column equilibrated with 50 mM Tris-$HNO_3$ buffer (150 mM $NaNO_3$, pH = 7.4). The identity of purified 6PGDH, RpoA, and PGL were further confirmed by MALDI-TOF MS. Plasmids for $6PGDH^{C168S}$, $6PGDH^{C330S}$, $6PGDH^{C363S}$, $6PGDH^{M357S}$, $6PGDH^{M140S}$, $6PGDH^{H185S}$, $6PGDH^{M239S}$, and $6PGDH^{H211S}$ mutants were constructed by site-directed mutagenesis using Phusion high fidelity DNA polymerase (NEB). WT pHisSUMO-*gnd* plasmid was used as a DNA template. The primers used for mutants are listed in Supplementary Table 13. The condition for expression and purification of 6PGDH mutants was identical to WT 6PGDH.

**GE-ICP-MS of purified proteins**. To measure the silver-binding capability of the recombinant proteins, typical one-dimensional GE-ICP-MS was used. For each GE-ICP-MS test, 10 µL of 4 µM proteins were loaded. $^{127}$I-labeled proteins were used as internal standards to calibrate MWs and intensity of Ag contained in proteins. A freshly prepared column gel, 2.5 cm 12% resolving gel and 0.6 cm 4% stacking gel were used for this study. All the purified proteins were subjected to the same separation conditions.

**MALDI-TOF mass spectrometry of purified proteins**. The binding stoichiometry of 6PGDH, RpoA, and Pgl to silver were measured by MALDI-TOF MS (Bruker ultraflex extreme MALDI-TOF-TOF-MS). Proteins after incubation with different molar equivalents of $Ag^+$ at room temperature for 1 h were mixed with saturated sinapic acid matrix (in ACN:$H_2O$ = 1:1). Prior to MALDI-TOF MS analysis, 1 µL proteins (10 µM) was mixed with 1 µL matrix and crystalized on polished 384-well plate. Mass spectra were measured in the positive linear mode.

**Isothermal titration calorimetry**. The RpoA, Pgl, 6PGDH were prepared in a Tris buffer (35 mM Tris-$HNO_3$, 100 mM $NaNO_3$, pH = 7.4) with a final concentration of 20 µM. The titrant is $AgNO_3$ at around 600 µM in Tris-$HNO_3$ buffer. In general, 40 µL of $AgNO_3$ were titrated into 200 µL proteins at 90 s intervals. The signals of the titration of $AgNO_3$ into the Tris-$HNO_3$ buffer were recorded as the background. All isothermal titration calorimetry (ITC) experiments were performed on a Malvern MicroCal ITC200 at 25 °C and all data were analyzed using the Origin software

**Enzyme inhibition assay for purified 6PGDH**. Freshly prepared 10 µM of 6PGDH, $6PGDH^{C168S}$, $6PGDH^{C330S}$, $6PGDH^{C363S}$, $6PGDH^{M357S}$, $6PGDH^{M140S}$, $6PGDH^{H185S}$, $6PGDH^{M239S}$, and $6PGDH^{H211S}$ in Tris-$HNO_3$ buffer (35 mM Tris-$HNO_3$ buffer, 100 mM $NaNO_3$, pH = 7.4) were incubated with various concentrations of $Ag^+$ for 1 h at room temperature. The enzymatic activities were measured according to previous reports[55]. Enzyme activity was determined by measuring the forward reaction rate of substrate 6PG oxidation and decarboxylation, in which the absorbance of the product NADPH was monitored at 340 nm. A typical reaction mixture contained 35 mM Tris-$HNO_3$ buffer (pH = 7.4), 10 mM $MgCl_2$, 0.5 mM NADP$^+$, and 2 mM 6PG. All enzyme assays were carried out at 25 °C in a final volume of 100 µL. The reaction was initiated by adding the enzyme to a final concentration of 10 nM. The absorbance at 340 nm was monitored constantly for a duration of 20 min. The initial rates were calculated from each reaction to fit the $IC_{50}$ curves.

For the enzyme kinetics, freshly prepared 10 µM of 6PGDH were incubated with 10, 20, and 40 µM $Ag^+$ for 1 h at room temperature. The enzymatic activity was measured with 6-phosphoglucoginic acid and NADP$^+$ as the substrate ranging from 0.05 to 1 mM. Control experiments were carried out without supplementation of $Ag^+$ under the same conditions. The catalytic activities of 6PGDH were assayed by measuring the absorbance of produced NADPH at 340 nm as described above. The assays were initiated by the addition of the enzyme. The $K_m$ and $V_{max}$ for both the uninhibited and inhibited reactions were obtained by fitting the data into the double reciprocal Lineweaver-Burk plots.

**X-ray crystallography**. Crystals of apo-6PGDH were grown using hanging-drop vapor diffusion method. The precipitant contains 0.1 M $NaNO_3$, 0.2 M $NH_4NO_3$, and 22% PEG 3350 (w/v). Tetragonal crystals appeared within 3 days and grew up to full size within 1 week. The crystals were soaked in cryo-protectant solution (0.1 M $NaNO_3$, 0.2 M $NH_4NO_3$, 25% PEG 3350 (w/v), 0.5 mM $AgNO_3$ and 20% glycerol) for 2 h before cryo-cooled. The diffraction data were collected at BL17U beamline of Shanghai Synchrotron Radiation Facility (SSRF), by using 0.97914 Å synchrotron radiation. The diffraction data were reduced with XDS[75]. The CCP4 suite[76] and Phenix[53] were used for data refinement and finalization. TLS refinement was used in the later stage of data processing. In brief, the structure of 6PGDH from *G. stearothermophilus* (PDB ID: 2W8Z) was used as the initial model for molecular replacement to solve the apo-form *Sa*6PGDH structure (PDB ID:7CB0). The 7CB0 was subsequently used as the model for molecular replacement of substrate-bound (PDB ID: 7CB5) and Ag-bound 6PGDH structures (PDB ID: 7CB6). The Ag-bound 6PGDH was resolved at 2.6 Å resolution with four polypeptide chains exist in each asymmetric unit. Identification of Ag was made on the significant positive peaks (≥20 σ) in the m*F*o-D*F*c (difference Fourier) map. The coordinates and structure factors for apo-, Ag-bound, and 6PG-bound 6PGDH were deposited at protein databank with accessing codes of 7CB0, 7CB6, and 7CB5, respectively.

**Evaluation of mutation frequencies and selection effects of antibacterial agents**. The mutation frequencies and selection effects of antibiotics and $Ag^+$/AgNP towards bacteria were evaluated via inoculation of sufficient bacteria, typically $10^6$ to $10^{10}$, on nutrient agar plate supplement with 2× MIC concentration of various antibiotics or $Ag^+$/AgNP. PVP-coated AgNP with a diameter of 10 nm was used, which was purchased from nanoComposix. AgNPs were synthesized by the borohydride reduction of $AgNO_3$ in the presence of PVP (40 kDa) as a stabilizing agent as previously reported[61,77]. The AgNPs were unagglomerated and monodispersed with spherical shape as evidenced by the characterization via transmission electron microscopy, UV-Vis, and dynamic light scattering by the manufacture. The CFU number of the inoculations were evaluated via serial agar dilution. The plates were incubated at 37 °C under aerobic condition for overnight. The mutation frequencies were calculated as dividing the evolved resistant colony number by CFU of inoculation. When no colony grown, the number of colonies considered as less than 1. The mutation frequency was calculated with number 1, and then presented as less than the calculated value. The percentage of resistant colony evolved in multiple separated tests was calculated to indicate the selection effects.

**Selection for silver resistance**. The silver resistance selection assay was performed with modification from the methods described previously[57]. *S. aureus* strain Newman were exposed to subinhibitory concentrations of $AgNO_3$ or AgNP repeatedly for 16 successive passages in microplates and BHI medium. AgNP/$AgNO_3$ (initial concentration of 256 μg/mL) were diluted with in a geometric progression in BHI medium and inoculated with *S. aureus* strain ($10^6$ CFU/mL). *S. aureus* cells were incubated with $AgNO_3$ or AgNP at 37 °C for 24 h in each step. Cultures of surviving bacteria were taken from the first three wells that contain subinhibitory concentrations of silver and followed by 1:1000 dilution into fresh medium supplemented with the same concentrations of $AgNO_3$ or AgNP. The MICs of $AgNO_3$ or AgNP were determined as the lowest silver concentration that inhibited visible growth of *S. aureus* after each incubation.

**Combination assay**. The MICs of each drug alone or in combination were determined by a typical broth microdilution method in accordance with CLSI standards[78]. In brief, *S. aureus* was cultured in BHI broth at 37 °C and 200 rpm for overnight. The bacterial density was adjusted to $1 \times 10^6$ CFU/mL and further confirmed by counting CFU on agar plates. Tested antibiotics (ampicillin, kanamycin, tetracycline, and ciprofloxacin) or $Ag^+$/AgNP or their combinations were added into 96-well plates with 2-fold serial dilution of the drugs, followed by addition of prepared bacterial inocula and incubation at 37 °C for overnight. Growth control wells containing the medium were included in each plate. The MIC was determined as the lowest concentration of the drug that could inhibit the growth of *S. aureus* by measuring the $OD_{600}$ via plate reader.

The FICI was calculated with the following equation: $FICI = FIC_A + FIC_B = C_A/MIC_A + C_B/MIC_B$, in which $MIC_A$ and $MIC_B$ are the individual MIC values of drug A or B, and $C_A$ and $C_B$ are the concentrations of drug A and B at the effective combinations. The FICI values were interpreted as follows: $FICI \leq 0.5$, synergy; $0.5 < FICI < 1$, partial synergy; $1 < FICI < 2$, additive; $2 < FICI < 4$, indifferent; $FICI > 4$, antagonism[79,80]. All tests were performed in three biological replicates.

**Time–kill kinetics assay**. The time–kill kinetics assays were preformed based on a standard method[81]. *S. aureus* cells were cultured aerobically in BHI medium at 37 °C and 200 rpm for overnight, and then diluted 100 times in fresh medium and cultured for another 2–3 h until they reach a log phase. The bacteria density was then adjusted to ~ $2 \times 10^7$ CFU/mL and followed by treatment of ampicillin (2 μg/mL), $Ag^+$ (8 μg/mL) or combined therapy. *S. aureus* cells without treatment were used as control. 0.1 mL aliquots of bacterial cultures were transferred for counting bacterial viability by agar plating at different time intervals. The results from three independent experiments were averaged and plotted as $\log_{10}$CFU/mL versus time (h).

**Resistance studies**. For single step resistance, the mutation prevention concentration (MPC) assay was preformed based on a modified method[82]. In brief, *S. aureus* Newman at $10^{10}$ CFU was plated onto BHI agar plate containing ampicillin and $Ag^+$/AgNP with various concentrations. After incubation for 48 h at 37 °C, the resistant mutant colonies were counted, and the relative mutation frequency was calculated as the proportion of resistant colonies per inoculum. For plates with observable colonies (>4), the colonies were re-cultured for MIC measurement. The MIC values higher than the original value was determined as higher-level resistant mutant colony. The concentrations that inhibited the growth of mutant colonies was defined as MPC. The multiple prevention index (MPI) was obtained via the equation of MPI = MPC/MIC.

For resistance development by sequential passaging, the assay was performed with modification from the previous method[20,83]. Generally, *S. aureus* Newman cells at exponential phase were diluted to ~$10^7$ CFU/mL in BHI broth. The diluted cell suspension was added to 96-well plate supplemented with drugs at escalating concentrations and incubated at 37 °C for 24 h prior testing the growth. For further passaging, cultures from the second highest concentrations allowing bacterial growth were performed 1:1000 dilution into fresh medium supplemented with the same concentrations of drugs. For ampicillin, 1-fold of MIC was set as 4 μg/mL. For the combination of ampicillin and $Ag^+$, 1-fold MIC was set as 0.25 μg/mL ampicillin + 8 μg/mL $Ag^+$. This in vitro passage was repeated for 16 times. The MIC was determined by broth microdilution every two passages. Experiments were performed with biological replicates.

**Statistical analysis**. Three biological replicates and two technique replicates were performed for all experiments without specified. All comparisons between two groups were performed on two-tailed *t*-test. Results are presented as mean ± SEM. Identification results of $Ag^+$-binding proteins are included in the Supplementary information.

**Reporting summary**. Further information on research design is available in the Nature Research Reporting Summary linked to this article.

## Data availability

The coordinates and structure factors for apo-6PGDH, 6PG-bound 6PGDH, and Ag-bound 6PGDH were deposited at protein databank with accessing codes of 7CB0, 7CB5, and 7CB6, respectively. The mass spectrometry proteomics data have been deposited to the ProteomeXchange Consortium via the PRIDE partner repository with the dataset identifier PXD025106. A reporting summary for this Article is available as a Supplementary Information file. All data supporting the findings of this study are available from the corresponding author upon reasonable request. Source data are provided with this paper.

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

## Acknowledgements

We thank the Research Grants Council of Hong Kong (17307017P, R7070-18, and F-HKU704/19), National Science Foundation of China (21671203) and the Norman & Cecilia Yip Foundation (the University of Hong Kong). We thank the Center for Genomic Sciences, Li Ka Shing Faculty of Medicine for the mass spectrometry facilities. We thank the helpful suggestions from Prof. K.M. Ng, Prof. Quan Hao, Prof. Wei Xia, Prof. Ligang Hu, and Dr. Yuchuan Wang. The crystal diffraction data were collected at Shanghai Synchrotron Radiation Facility (SSRF), the Chinese Academy of Sciences. We thank the staff at BL17U1 beamline for their kind help.

## Author contributions

H.W., H.S., and H.L. designed the study and wrote the manuscript. H.W. performed the study of protein separation, identification, validation, and bioinformatics analysis. H.W., P.G., Z.X., X.X., and R.Y.K. performed the experiments of mechanism study and analyzed data. H.W. and X.X. purified the proteins and carried out the biophysical characterizations. H.W. performed the protein crystallization, and H.W. and Q.Z. collected crystal diffraction data. M.W., H.W., and A.Y. resolved the crystal structures. H.W. performed all the cell-based tests. All authors commented the manuscript.

## Competing interests

The authors declare no competing interests.
