## [Peer Review File. · Nature Communications]

REVIEWER COMMENTS

Reviewer #1 (Remarks to the Author):

The authors around Sun et al present here a work of elucidating the molecular targets of silver ions in *S. aureus*, one of the most dangerous pathogens in e.g. hospitals, including its multi-resistant version. Silver has been known to have a lot of targets in bacteria, hence its excellent activity to kill or hamper the growth of bacteria in general.

Now, authors claim to do a first analysis of molecular targets for this specific bacterial strain. This is probably correct, but it needs to be mentioned that a similar, but maybe not as complete study was made for *E. coli* previously, showing up- and down-regulation of proteins (DOI: 10.1128/AAC.01830-09). Also, it should be said that (Gram negative) bacteria can get resistant to bacteria (at least to a certain extent) using the silver or copper efflux pumps (see <https://pubs.rsc.org/en/content/articlelanding/2018/cc/c8cc03784a#!divAbstract>). Protein-silver interactions have been studied as well and show preferential binding to his and met amino acids, as also discussed in this manuscript. Further effects were recently described in a review (<https://doi.org/10.1002/ppsc.201900419>). Also, the synergic effect of silver and antibiotics has been studied before, e.g. <http://onlinelibrary.wiley.com/doi/10.1002/cmdc.201400072/abstract>.

Hence, I would suggest to add these references and a corresponding discussion.

The discussion would also profit from mentioning the biocompatibility (or not) of silver with the human body.

Furthermore, the findings are important and likely more precise than previous analyses, but to a rather limited community, hence I would recommend journals like *Antimicrobial agents and chemotherapy* or similar calibers.

Reviewer #2 (Remarks to the Author):

The manuscript contains a very comprehensive metalloproteomic study of the toxicity of Ag⁺ to *Staphylococcus aureus*. The authors have identified specific Ag binding proteins and have combined their results with bioinformatics analysis to interpret the mechanism behind this well known antibacterial activity. Furthermore the authors have obtained structural information on the Ag⁺ binding and inhibitory effect on one of the enzymes uncovered in this study. And finally they determined synergistic effects of Ag/AgNP treatment and antibiotics to combat multiple resistant *S. aureus* strains.

The study is highly interesting and provides a breakthrough by comprehensively identifying the effects of Ag on the bacterial proteome. The antimicrobial activity of Ag⁺ and AgNP has been known for a long time, and despite considerable research efforts the molecular mechanisms have not been well described. This study offers an important advancement in our molecular understanding of the antibacterial activity of Ag⁺, which is important for the development of effective treatments against resistant bacteria.

The most difficult part of the study is the interpretation of the combined metalloproteomic and bioinformatic results. My main concern involves the suggested "divergence of the metabolic pathway towards the oxPPP pathway". The evidence is not completely convincing regarding this point (see more detailed comments below). The manuscript is acceptable for publication in Nature Communications after revision, as suggested below.

Detailed comments:

line 19: replace "to the technique challenge" by "to technical challenges". In my opinion the main challenge is the reactivity of Ag⁺ itself, e.g. with Cl⁻ and other medium components.

line 20: replace "home-made" by "in-house developed hyphenated technique LC-GE-ICP-MS"

line 26: "morphing the catalytic pocket..". What does this mean exactly?

line 46-47: "but a barren of antibiotic development pipeline nowadays," please rephrase as this sentence is not very clear.

line 56: "resistant breakers". resistant or resistance?

line 57: "Silver (Ag)". It is very important to clearly describe metallic silver or silver ions (Ag⁺). For most part of the manuscript Ag⁺ is used, which I recommend to be introduced already here.

line 58: remove "nowadays"

line 59: remove "have been devoted" and add "research" between "enormous" and "efforts"

line 59: "silver". Ag⁺ or metallic silver?

line 62: replace "no silver direct binding protein targets" by "no proteins that directly bind Ag+..."

line 63-64: I think it is mostly hindered by the chemistry of Ag⁺ itself as it is insoluble in combination with Cl⁻

line 66: the introduction of the existing metalloproteomic techniques and why these are not suitable is not comprehensively described. How did the authors solve the problems of Ag⁺ reactivity with non-protein medium or buffer components?

line 79: Although it is fine to use a particular measure to find a suitable silver concentration for the metalloproteomics, the study lacks the effect of silver on the growth rate of *Sau*. Are there effects on the lag phase or on the maximum specific growth rate? What happened to the growth of *Sau* at 0.12 mM Ag⁺ in BHI?

line 80: 20 mg/L is 0.12 mM silver nitrate. This is quite a high concentration. Could it be that the complex medium reduces the concentration of Ag significantly? It has been reported that medium components greatly influence the effective antimicrobial concentration of Ag⁺. See: <https://doi.org/10.1038/s41598-018-27540-9>

Is anything known about Ag⁺ toxic concentrations in BHI and minimal media?

line 102: Are these proteins also abundant proteins, or was the ¹⁰⁷Ag abundance (intensity) normalized for the protein abundance.

line 114: The interaction of Ag⁺ with three purified proteins was tested. Did the authors also use a non-binding protein as a control experiment? I can imagine that under the right conditions any protein will bind Ag⁺ (e.g. silver staining proteins).

line 128-129: Fig. 1A: What does growth 100% mean? Was this the effect of silver on the maximum specific growth rate (doubling time), or on the maximum number of cells or CFU obtained after a certain time. The concentration Ag⁺ in (micro/milli)molar is more informative than microgram/ml.

BHI is a very rich medium, part of the Ag⁺ will react with medium components, is there any idea how much free Ag⁺ is still available?

Fig. 1D: what are "others". is this the cell wall? *S. aureus* is a gram positive bacterium.

line 150: The authors should explain what the analysis with STRING exactly involves. The list of identified Ag⁺ binding proteins is not that long, so the authors could already discuss the identified

proteins without relying solely on the bioinformatics analysis. Are these proteins cys-rich? Ag⁺ is a soft acid and will likely react with accessible S(Cys) or S(Met), as the authors have found for 6PGDH.

line 154: "single-organism carbohydrate metabolic process". What does this mean?

line 157: "identified proteins are involved in...ribosome...". The involvement of the ribosomal proteins is in "translation" or "protein synthesis". Indeed a lot of ribosomal proteins are identified. The resulting effect Ag⁺ may have on protein synthesis is not that well discussed. It is interesting to realize many antibiotics for which resistance can develop are inhibitors of ribosomal function (e.g. chloramphenicol).

line 159: "intracellular and cytoplasm". What is the difference in *S. aureus*?

line 160: what does "macromolecular complex" mean here?

line 163: "silver peaks". Was the silver amount corrected for protein abundance, or could it reflect abundant proteins.

line 171: remove "to eradicate"

line 173: how do these two identified functional categories relate to the oxidative stress response discussed above?

line 182-183: The glycolytic enzymes are regulated by allosteric interactions and transcription and translation. Given the rather global impact on the cellular physiology is it possible to elucidate the effect on silver on a single enzyme by doing whole cell experiments? Could it be there is just less Pfk or Eno present due to indirect effects of Ag⁺? These indirect effects seem to be present for Ldh. Why not for the other enzymes?

line 188: 200 microM Ag⁺ is significantly higher than the MIC₅₀ at 120 microM. Why was this concentration chosen?

line 191: is the ATP abundance the intracellular ATP concentration or the ATP/ADP ratio? If I look at the SI methods it is the ATP concentration.

line 201: "revolutionized". What does this mean in this sentence? Please rephrase.

line 215-216: Was the ROS the cause or an effect of cell toxicity/death.

line 233: "in vivo activities of Pgl and 6PGDH". Were these in-vitro activity assays with whole cells, or truly in-vivo activities?

line 249: "induced a metabolic diverting". please rephrase. Maybe: "induced the diversion of the metabolic flux from glycolysis to oxPPP"

line 250: "The shift from glycolysis to oxPPP was not shown at the metabolite level? The enzymes are inhibited by Ag⁺ - is there independent evidence for this change of flux?

line 253 and Fig 2H: Here are the growthcurves of WT and the knockouts with and without silver presented. Please fit the lag phase, OD_{max} and maximum specific growthrates (u-max) on the basis of these figures. On the basis of these curves (and biological replicates) are there significant differences in the growth rate?

The primary effect seems to be on the lag-phase. I don't think the maximum specific growthrate is that much different for any of the Ag⁺ inhibited strains.

line 256-257: I don't follow this exactly. The knockout of two silver binding proteins makes it more sensitive to silver (apparently slower growth in the presence of Ag⁺)?

line 259: the reduction of the PPP enzyme activities is evidence against the increased flux through the PPP, despite the protein upregulation.

line 262: Is the metabolic redirection futile, or does it not happen at all? I'm not really convinced that at the start of the Ag⁺ interaction this increased flux through the oxPPP really happens to start with. As the authors state Ag⁺ disrupts this pathway. Maybe this is a semantic discussion, the upregulation of the proteins of oxPPP is futile because the enzymes are inhibited.

line 268: this part is a bit confusing, as mentioned in my previous comment. Consider rephrasing this part, or provide independent evidence that the redirection of metabolic flux towards oxPPP actually happened.

line 272: "induced the generation of ROS at the late stage". Or: "inhibits the suppression of ROS generation". It seems like the primary antioxidant mechanisms are disturbed by Ag⁺ which could lead to accumulation of ROS by other means.

Is ROS generation leading to this apparent change in metabolism?

line 273: "the activated oxPPP contributes to the alleviation of Ag⁺ toxicity to S.aureus shortly,...". Where is the evidence to support this? In Fig 2I the activity of the oxPPP enzymes did not increase at the start.

line 277: replace "home-made" by "in-house developed"

line 293-294: Would more Ag⁺ bind if you add more Ag equivalents?

line 295: molar ratio increases to 4, but 5 eq were added. is this correct?

line 298: "non-competitive inhibition". Ag⁺ is not an reversible inhibitor, I presume. The enzyme is chemically modified by Ag⁺, which apparently does not change the apparent K_M, indicating the substrate binding is spared, but the rate determining step is affected.

line 339: "Nam ring". nicotinamide?

line 391-392, Fig. 4C: The MIC values are very discrete. Identical for most passages with Ag and AgNP, or exactly 2x that value. Were the tested concentrations in large steps? Is it expected that the MIC of AgNP is exactly the same as for Ag⁺?

line 417: how toxic are these silver levels for human cells?

line 458-459: again, I'm not so sure about the metabolic divergence. See comments above.

SI, table 2: "Protein map" is not a very informative annotation. This protein is an "MHC class II analog protein"

Reviewer #3 (Remarks to the Author):

This is another very extensive comprehensive paper from this group, following up on their study of the Ag-proteome in E. coli, here applying their tool of LC-GE-ICP-MS to S. aureus.

concerns/changes/comments

- The primary discovery of the study is the identification of the S. aureus Ag-proteome. The authors add additional conformational validation experiments of structural biology and mutants which is good. However, I do not feel that the Ag-antibiotic synergy data fits in this study (lines 357 to 430). Although this work is good and important, it is less novel and it is lost and the findings are in this lost in the proteome study. Highlighted but lack of inclusion of specifics in the abstract. The removal of this section would allow the authors to elaborate more on their other data.

-For antibiotic resistance different strains and the media can give rise to dramatic differences. Although such information is included in methods, on line 79 the strain number of the S. aureus used should be included as well as the media used for these experiments. This is also important as different media will lead to different proteome present in the cell. A recognition of this is important that one has not sampled the entire proteome.

- As some good protein chemistry and enzymology are done on some of the targets. It would be good to complement the Ag binding to RpoA Pgl and Gnd by ITC that will give affinity and validate the stoichiometries by ICP-MS.

- I found it a bit vague for the location of Ag-proteins found in figure 1D as 'others' seems only two possible categories, soluble or membrane, what is meant by other location?

- I was confused about the enrichment figure Fig 1 I, the text indicates Glycolytic processes were a major target yet the bar on the curve is the smallest with an intermediate FDR number. This is counter-intuitive and thus the authors should clarify in the text to explain why they feel glycolytic is more important than say nitrogen related metabolism with longer bars.

- Section on oxidative stress and ROS, lines 197...; I think these findings are very important to define how Ag exposure leads to increased ROS. The literature is full of conflicting reports of Ag and ROS production. As Ag is not redox active it is highly unlikely that it produces ROS through Fenton like reactions. Her authors identify poisoning of the oxidative 'repair' system, which means ROS levels would increase as the repair is not functioning. NOT because Ag generates ROS. This should be stated more clearly. Also in figure 4A, it is cartooned that the Ag ions are producing ROS, which is

incorrect and we must stop propagating this misinterpretation, as is nicely found from the present study, it is the poisoning of the proteins that remove ROS that leads to increased cell levels of ROS.

Also, line 201 wording is awkward; 'revolutionized by S.aureus to regulate the oxidative stress' should be more like 'utilized by S. a. to respond to oxidative stress'

- Figure display suggestion. In Fig2 A, D, F the Ag atoms are depicted as grey balls, which is the same coloration as the enzymes in the pathway and thus difficult to notice right away. in Figure 4A the Ag atoms are red balls, be consistent and use red balls in figure 2 as well. in Figure 3 structure the Ag atoms as grey balls work ok. Also, the pathway enzymes in A are abbreviated and F full names are given. Please be consistent.

Reviewer #4 (Remarks to the Author):

The manuscript entitled "Multi-target mode of action of silver against *Staphylococcus aureus* endows it with capability to combat antibiotic resistance" authored by H. Wang et al. presents a broad study applied to understand the mode of action of silver against the antibiotic-resistant bacteria *Staphylococcus aureus*.

The study uses a Liquid-Chromatography-Gel electrophoresis-Inductively Coupled Plasma Mass Spectrometry system to separate and identify 38 Ag⁺ binding proteins from different metabolic pathways. Besides, bioinformatic analysis and biochemical validations were used to demonstrate that silver targets multiple proteins, thus interfering with multiple pathways, including glycolysis, oxidative Pentose Phosphate Pathway (oxPPP), and Reactive Oxygen Species stress defense system, to exert its bactericidal effect against *S. aureus*. Furthermore, Ag⁺ ions presence was demonstrated in the crystallographic structure of 6-phosphogluconate dehydrogenase of the oxidative pentose phosphate pathway.

Overall the study provides insights into the action of silver ions on antibiotic resistant *S aureus*. However, I am not convinced that the advances provided for some of the experiments are of sufficient quality to merit publication in Nature communications.

I have some comments that the authors may want to consider to improve the quality of the manuscript:

- 1.The title announces the capability of silver to combat antibiotic-resistance in *Staphylococcus aureus*; however, the properties of silver and silver nanoparticles applicable to human treatments are still under investigation in laboratory and animal studies. Therefore, how can then be established that these ions combat antibiotic-resistance, in what context? How can the inhibition of human proteins be prevented?

2. There is no methodology included in the main body of the manuscript. This section is a central part of the Supplementary Methods. Definitively, there should be a Methodology section with the most relevant experiments adequately explained.

3. There is no explanation of how the crystal structures of 6D-phosphogluconate dehydrogenase, in the absence and presence of silver ions, were solved. There is a few-lines section (X-ray crystallography) in Supplementary material that, however, does not explain how the phase problem was solved for both structures (7CB6 and 7CB5). The structure's description is also incomplete; there is no indication of how many polypeptide chains exist in the asymmetric unit and if the interactions of Ag⁺ is identical. Besides, it is not explained why the apoenzyme was used for these experiments.

4. From MALDI-TOF experiments, five Ag⁺ ions bound to the enzyme 6PGDH were detected; however, no electron density map is included in Fig. 3 to judge the validity of the coordinates for ions and substrate bound to the enzyme. The author should show non-biased electron density maps for ligands.

5. Two preliminary validation reports are attached. The final validation reports for both structures should have been included for the revision of this manuscript.

6. Supplementary Table 9 (Summary of X-ray crystallography data collection and refinement) is incomplete. CC1/2 values, Rmerge or Rmeas values, redundancies, and data for the highest resolution shell are missing. B factors for all atoms should also be included.

Leave out redundant cell angles for space group 19.

7. In Section "Multi-target mode of action of silver against *S. aureus* suppresses its antibiotic selection effects" and Supp. Material sections "Evaluation of mutation frequencies and selection effects of antibacterial agents" and "Combination assay", and "Resistance studies", Ag nanoparticles were used; however, there is no description of what kind of nanoparticles the authors used for these experiments, or how these AgNPs were prepared.

8. In some cases, there is no correspondence on what it is said in the main document with Supplementary material. For instance, it is said: "The corresponding peaks were fractionized, collected, and subjected to MALDI-TOF-MS analysis and identified through peptide mass fingerprinting after further purification by SDS-PAGE (Supplementary Fig. 3). However, the caption to Supplementary Fig. 3 is "GE-ICP-MS profile of 107Ag corresponding to Ag-415 bound RpoB at MW of 135.9 kDa.

Minor:

Fig. 3(H) indicates residue F238, it should say M239.

English language revision should be performed, there are minor problems. For instance, the title: Multi-target mode of action of silver against *Staphylococcus aureus* endows it with the capability to combat antibiotic resistance.

TEL: (852) 2859 8974

FAX: (852) 2857 1586

E-mail: hsun@hku.hk

Prof. Hongzhe Sun

Norman & Cecilia Professor in Bioinorganic Chemistry

DEPARTMENT OF CHEMISTRY
THE UNIVERSITY OF HONG KONG
POKFULAM ROAD, HONG KONG

Re: “Multi-target mode of action of silver against *Staphylococcus aureus* enables its potential for combating antibiotic resistance” (NCOMMS-20-35808)

REVIEWER COMMENTS

Reviewer #1 (Remarks to the Author):

The authors around Sun et al present here a work of elucidating the molecular targets of silver ions in *S. aureus*, one of the most dangerous pathogens in e.g. hospitals, including its multi-resistant version. Silver has been known to have a lot of targets in bacteria, hence its excellent activity to kill or hamper the growth of bacteria in general.

Now, authors claim to do a first analysis of molecular targets for this specific bacterial strain. This is probably correct, but it needs to be mentioned that a similar, but maybe not as complete study was made for *E. coli* previously, showing up- and down-regulation of proteins (DOI: 10.1128/AAC.01830-09). Also, it should be said that (Gram negative) bacteria can get resistant to bacteria (at least to a certain extent) using the silver or copper efflux pumps (see <https://pubs.rsc.org/en/content/articlelanding/2018/cc/c8cc03784a#divAbstract>). Protein-silver interactions have been studied as well and show preferential binding to his and met amino acids, as also discussed in this manuscript. Further effects were recently described in a review (<https://doi.org/10.1002/ppsc.201900419>). Also, the synergic effect of silver and antibiotics has been studied before, e.g. <http://onlinelibrary.wiley.com/doi/10.1002/cmdc.201400072/abstract>.

Hence, I would suggest to add these references and a corresponding discussion.

The discussion would also profit from mentioning the biocompatibility (or not) of silver with the human body. Furthermore, the findings are important and likely more precise than previous analyses, but to a rather limited community, hence I would recommend journals like *Antimicrobial agents and chemotherapy* or similar calibers.

[General Response to Reviewer #1]

We truly thank the reviewer's helpful suggestions, which are immensely useful for us to improve the manuscript. We have added a discussion section (highlighted in Page 17-18) in the revised manuscript, in which the references mentioned by the reviewer are cited. In the discussion part, we clearly elaborated the major merits of our study and clarified how this study being distinct from previous ones to advance our understanding on the mode of action of Ag⁺ against *S. aureus*. Given the novel discovery and the solid evidence to support the conclusions, we believe that this study deserves its publication in *Nature Communications*.

TEL: (852) 2859 8974
FAX: (852) 2857 1586
E-mail: hsun@hku.hk

Prof. Hongzhe Sun
Norman & Cecilia Professor in Bioinorganic Chemistry

DEPARTMENT OF CHEMISTRY
THE UNIVERSITY OF HONG KONG
POKFULAM ROAD, HONG KONG

Reviewer #2 (Remarks to the Author):

The manuscript contains a very comprehensive metalloproteomic study of the toxicity of Ag⁺ to *Staphylococcus aureus*. The authors have identified specific Ag binding proteins and have combined their results with bioinformatics analysis to interpret the mechanism behind this well known antibacterial activity. Furthermore the authors have obtained structural information on the Ag⁺ binding and inhibitory effect on one of the enzymes uncovered in this study. And finally they determined synergistic effects of Ag/AgNP treatment and antibiotics to combat multiple resistant *S. aureus* strains.

The study is highly interesting and provides a breakthrough by comprehensively identifying the effects of Ag on the bacterial proteome. The antimicrobial activity of Ag⁺ and AgNP has been known for a long time, and despite considerable research efforts the molecular mechanisms have not been well described. This study offers an important advancement in our molecular understanding of the antibacterial activity of Ag⁺, which is important for the development of effective treatments against resistant bacteria.

The most difficult part of the study is the interpretation of the combined metalloproteomic and bioinformatic results. My main concern involves the suggested "divergence of the metabolic pathway towards the oxPPP pathway". The evidence is not completely convincing regarding this point (see more detailed comments below). The manuscript is acceptable for publication in *Nature Communications* after revision, as suggested below.

[General Response to Reviewer #2]

We truly appreciate the reviewer's favorable comments and helpful suggestions on the present study as well as recognition on the major merits of our work. These suggestions are immensely useful for us to improve the manuscript.

Detailed comments:

line 19: replace "to the technique challenge" by "to technical challenges". In my opinion the main challenge is the reactivity of Ag⁺ itself, e.g. with Cl⁻ and other medium components.

[Response] Agreed and revised accordingly.

TEL: (852) 2859 8974
FAX: (852) 2857 1586
E-mail: hsun@hku.hk

Prof. Hongzhe Sun
Norman & Cecilia Professor in Bioinorganic Chemistry

DEPARTMENT OF CHEMISTRY
THE UNIVERSITY OF HONG KONG
POKFULAM ROAD, HONG KONG

line 20: replace "home-made" by "in-house developed hyphenated technique LC-GE-ICP-MS"

[Response] Agreed and replaced accordingly.

line 26: "morphing the catalytic pocket..". What does this mean exactly?

[Response] "Morphing the catalytic pocket" means "The shape of the 6PGDH catalytic pocket has changed due to the binding of four Ag⁺ ions to the residues that are close to the pocket". We have added the explanation into the revised manuscript as suggested.

line 46-47: "but a barren of antibiotic development pipeline nowadays," please rephrase as this sentence is not very clear.

[Response] We have rephrased it to "but a lack of antibiotic development pipeline nowadays".

line 56: "resistant breakers". resistant or resistance?

[Response] Agreed and rephrased to "resistance breakers".

line 57: "Silver (Ag)". It is very important to clearly describe metallic silver or silver ions (Ag⁺). For most part of the manuscript Ag⁺ is used, which I recommend to be introduced already here.

[Response] Agreed and revised to "Silver ions (Ag⁺)" accordingly.

line 58: remove "nowadays"

[Response] Removed.

line 59: remove "have been devoted" and add "research" between "enormous" and "efforts"

[Response] Revised accordingly.

line 59: "silver". Ag⁺ or metallic silver?

[Response] Revised to "Ag⁺".

line 62: replace "no silver direct binding protein targets" by "no proteins that directly bind Ag+..."

[Response] Agreed and replaced as suggested.

TEL: (852) 2859 8974
 FAX: (852) 2857 1586
 E-mail: hsun@hku.hk

Prof. Hongzhe Sun
Norman & Cecilia Professor in Bioinorganic Chemistry

DEPARTMENT OF CHEMISTRY
 THE UNIVERSITY OF HONG KONG
 POKFULAM ROAD, HONG KONG

line 63-64: I think it is mostly hindered by the chemistry of Ag⁺ itself as it is insoluble in combination with Cl-
[Response] Agreed. “The chemistry of Ag⁺” was added.

line 66: the introduction of the existing metalloproteomic techniques and why these are not suitable is not comprehensively described. How did the authors solve the problems of Ag⁺ reactivity with non-protein medium or buffer components?

[Response] As suggested, we have added more description of the existing metalloproteomic techniques, such as LA-ICP-MS, LC-ICP-MS, XRF, and fluorescent probes, and comprehensively described their limitations in the introduction of revised manuscript (page 2, line 65-70).

For buffer, we used the system composed of 35 mM Tris-HNO₃, 100 mM NaNO₃-HNO₃ (pH 7.4) to avoid precipitation of silver ions (as AgCl).

line 79: Although it is fine to use a particular measure to find a suitable silver concentration for the metalloproteomics, the study lacks the effect of silver on the growth rate of *Sau*. Are there effects on the lag phase or on the maximum specific growth rate? What happened to the growth of *Sau* at 0.12 mM Ag⁺ in BHI?

[Response] We truly appreciate the reviewer’s helpful suggestion. The effect of Ag⁺ on the growth rate of *S. aureus Newman* in BHI was evaluated by measuring the OD₆₀₀. As shown in Figure 1, addition 118 μM (20 μg/mL) Ag⁺ led to longer lag phase ($\lambda = 2.79$ h) and a decrease of maximum specific growth rate ($\mu_{max} = 0.09$ OD/h) than the untreated *S. aureus* cells ($\lambda = 0.90$ h and $\mu_{max} = 0.24$ OD/h).

We have incorporated the figure and description into the revised manuscript accordingly (page 3, line 86-88 and Fig. 1B).

Figure 1. The growth curves of *S. aureus Newman* after treatment of various concentrations of Ag⁺. The results are shown as mean ± SEM (n = 3).

TEL: (852) 2859 8974
 FAX: (852) 2857 1586
 E-mail: hsun@hku.hk

Prof. Hongzhe Sun
Norman & Cecilia Professor in Bioinorganic Chemistry

DEPARTMENT OF CHEMISTRY
 THE UNIVERSITY OF HONG KONG
 POKFULAM ROAD, HONG KONG

line 80: 20 mg/L is 0.12 mM silver nitrate. This is quite a high concentration. Could it be that the complex medium reduces the concentration of Ag significantly? It has been reported that medium components greatly influence the effective antimicrobial concentration of Ag⁺. See: <https://doi.org/10.1038/s41598-018-27540-9>

Is anything known about Ag⁺ toxic concentrations in BHI and minimal media?

[Response] We truly appreciate the reviewer's insightful comments and agree that the medium components could significantly influence the effective antimicrobial concentration of Ag⁺. The measured MIC (20 µg/mL) of AgNO₃ against *S. aureus Newman* in BHI is generally in good agreement with previous reports (8 - 32.5 µg/mL) in rich medium (*J Antimicrob Chemother*, 2013, 68, 131-138; *RSC Adv*, 2018, 8, 20829-20835; *Chem Sci*, 2017, 8, 8061-8066).

As suggested, we further measured the effective concentration of Ag⁺ in BHI by ICP-MS. As shown in Figure 2, the effective concentration is 7.6 µg/mL (45 µM) when 20 µg/mL (118 µM) AgNO₃ is added into the BHI medium.

Figure 2. Effective concentration of Ag⁺ in BHI medium as measured by ICP-MS (n = 3). The results are shown as mean ± SEM.

line 102: Are these proteins also abundant proteins, or was the ¹⁰⁷Ag abundance (intensity) normalized for the protein abundance.

[Response] In Fig. 1F, the y-axis denotes the ¹⁰⁷Ag intensity, which is the integration of peaks corresponding to each Ag⁺-binding protein and not normalized to protein abundance since it is very difficult to accurately quantify the abundance of individual protein.

line 114: The interaction of Ag⁺ with three purified proteins was tested. Did the authors also use a non-

TEL: (852) 2859 8974
 FAX: (852) 2857 1586
 E-mail: hsun@hku.hk

Prof. Hongzhe Sun
Norman & Cecilia Professor in Bioinorganic Chemistry

DEPARTMENT OF CHEMISTRY
 THE UNIVERSITY OF HONG KONG
 POKFULAM ROAD, HONG KONG

binding protein as a control experiment? I can imagine that under the right conditions any protein will bind Ag⁺ (e.g. silver staining proteins).

[Response] We truly appreciate the suggestion and understand the concern from the reviewer. As suggested, we added a control experiment by using *S. aureus* glutamyl endopeptidase (GEP) protein (29 kDa), which is not identified as Ag⁺ binding protein by our method. GEP protein was incubated with 5 eq. of Ag⁺ overnight and then subjected to GE-ICP-MS analysis. As shown in Figure 3, no ¹⁰⁷Ag peak in GE-ICP-MS profile at the MW *ca.* 30 kDa was observed, demonstrating that our method could precisely track Ag⁺-binding proteins under physiological relevant conditions.

We agree with the reviewer that many proteins could bind to Ag⁺ under some specific condition, such as silver staining. However, silver staining is based on the selective reduction of silver ions into metallic silver by aldehyde and the staining process sequentially consists of protein fixation, sensitization, washing, silver impregnation, and development for imaging. All these procedures are not physiologically relevant. In the current study, we focus on identification of Ag⁺-binding proteins in *S. aureus* under physiological conditions. We admit that our technique is not perfect. Proteins with low Ag⁺-binding affinity might be dissociated during the gel electrophoresis separation. Nevertheless, the method appears to be the only technique enabling tracking Ag⁺ proteome *in vivo* precisely at current stage.

The experimental results and discussion are added into the revised manuscript as suggested (page 4, line 124-127 and supplementary Fig. 3B).

Figure 3. GE-ICP-MS profile of purified GEP protein incubated with 5 eq. of Ag⁺.

line 128-129: Fig. 1A: What does growth 100% mean? Was this the effect of silver on the maximum specific growth rate (doubling time), or on the maximum number of cells or CFU obtained after a certain time. The concentration Ag⁺ in (micro/milli)molar is more informative than microgram/ml.

TEL: (852) 2859 8974
FAX: (852) 2857 1586
E-mail: hsun@hku.hk

Prof. Hongzhe Sun
Norman & Cecilia Professor in Bioinorganic Chemistry

DEPARTMENT OF CHEMISTRY
THE UNIVERSITY OF HONG KONG
POKFULAM ROAD, HONG KONG

BHI is a very rich medium, part of the Ag⁺ will react with medium components, is there any idea how much free Ag⁺ is still available?

Fig. 1D: what are "others". is this the cell wall? *S. aureus* is a gram positive bacterium.

[Response] We thank the reviewer's careful reading of the manuscript. The OD₆₀₀ of *S. aureus* cells without treatment of AgNO₃ was set as 100%. The growth of *S. aureus* treated with different concentrations of AgNO₃ was normalized to that of the control group. The explanation was added to the figure legend.

-- As shown in Figure 1, the initial growth (AgNO₃ was added at the exponential growth phase) of *S. aureus* was time dependently delayed with the increase of Ag⁺ concentrations, which also result in an Ag⁺ dose-dependent decrease of maximum number of the cells.

-- We have shown the concentration in "μM" in revised manuscript

-- As shown in Figure 2, the effective concentration 7.6 μg/mL when 20 μg/mL AgNO₃ is supplemented the BHI media.

-- The "others" is the undissolved pellet by 1% SDS, which mainly composed by cell wall.

line 150: The authors should explain what the analysis with STRING exactly involves. The list of identified Ag⁺ binding proteins is not that long, so the authors could already discuss the identified proteins without relying solely on the bioinformatics analysis. Are these proteins cys-rich? Ag⁺ is a soft acid and will likely react with accessible S(Cys) or S(Met), as the authors have found for 6PGDH.

[Response] Agreed and an explanation of STRING analysis has been added.

-- Agreed. The number of cysteine ranges from 1 to 9 for the identified Ag⁺-binding proteins and all of them contain methionine and histidine.

line 154: "single-organism carbohydrate metabolic process". What does this mean?

[Response] The "single-organism carbohydrate metabolic process" (GO:0044723) means the chemical reactions and pathways involving carbohydrates, occurring within a single organism. We revised it as "carbohydrate metabolic process" to avoid misunderstanding.

line 157: "identified proteins are involved in...ribosome...". The involvement of the ribosomal proteins is in "translation" or "protein synthesis". Indeed a lot of ribosomal proteins are identified. The resulting effect Ag⁺ may have on protein synthesis is not that well discussed. It is interesting to realize many antibiotics for which resistance can develop are inhibitors of ribosomal function (e.g. chloramphenicol).

TEL: (852) 2859 8974
 FAX: (852) 2857 1586
 E-mail: hsun@hku.hk

Prof. Hongzhe Sun
Norman & Cecilia Professor in Bioinorganic Chemistry

DEPARTMENT OF CHEMISTRY
 THE UNIVERSITY OF HONG KONG
 POKFULAM ROAD, HONG KONG

[Response] Agreed. We further measured the effect of Ag^+ on nascent protein synthesis of *S. aureus* via O-propargyl-puromycin (OPP) labeling and Click-iT chemistry. As shown in Figure 4, the fluorescence signals from OPP which is incorporated into the newly synthesized peptides were decreased with the increase in Ag^+ concentration, demonstrating Ag^+ inhibits the nascent protein synthesis of *S. aureus* in a dose-dependent manner, which is consistent with our findings by metalloproteomic approach, *i.e.*, many ribosomal proteins are identified as Ag^+ binding proteins.

The experimental results and discussion are added into the revised manuscript as suggested (page 17, line 486-492 and supplementary Fig. 15).

Figure 4. The effect of Ag^+ on nascent protein synthesis of *S. aureus* ($n = 3$). The antibiotic of tetracycline, which is known to target 30S and disrupt protein synthesis, was used as a control. The fluorescence of *S. aureus* cells with cell number of 1 OD was recorded. The results are shown as mean \pm SEM.

line 159: "intracellular and cytoplasm". What is the difference in *S. aureus*?

[Response] Revised to "cytoplasm".

line 160: what does "macromolecular complex" mean here?

[Response] In the STRING analysis, the cellular component ontology describes locations of proteins, at the levels of subcellular structures and macromolecular complexes. "Macromolecular complex" means a stable set of (two or more) interacting proteins. Non-protein molecules (*e.g.*, small molecules, nucleic acids) may also be present in the complex if they are an integral part of the complex. These complexes carry out essential process in the cell.

TEL: (852) 2859 8974
FAX: (852) 2857 1586
E-mail: hsun@hku.hk

Prof. Hongzhe Sun
Norman & Cecilia Professor in Bioinorganic Chemistry

DEPARTMENT OF CHEMISTRY
THE UNIVERSITY OF HONG KONG
POKFULAM ROAD, HONG KONG

line 163: "silver peaks". Was the silver amount corrected for protein abundance, or could it reflect abundant proteins.

[Response] The integration of the ^{107}Ag peak of each Ag^+ -binding protein could reflect the silver content contained in the corresponding protein, which is not normalized to protein abundance since it is very difficult to accurately quantify individual protein. Higher peak means higher silver content contained in the corresponding protein, which might be ascribed to relatively higher abundance of these proteins or higher binding stoichiometry between the protein and silver.

line 171: remove "to eradicate"

[Response] Removed.

line 173: how do these two identified functional categories relate to the oxidative stress response discussed above?

[Response] The functional categories enriched indeed represent the biological pathway with highest number of proteins while the silver content contained in each biological pathway are the sum of silver content of proteins in each pathway. These two functional categories are not directly related to the oxidative stress response. To avoid misleading, we removed this sentence.

line 182-183: The glycolytic enzymes are regulated by allosteric interactions and transcription and translation. Given the rather global impact on the cellular physiology is it possible to elucidate the effect on silver on a single enzyme by doing whole cell experiments? Could it be there is just less Pfk or Eno present due to indirect effects of Ag^+ ? These indirect effects seem to be present for Ldh. Why not for the other enzymes?

[Response] We truly thank the reviewer's insightful comments. We agree with the reviewer that we need to consider the indirect effect from different expression levels of the proteins under various conditions. Thus, we performed the real time quantitative polymerase chain reaction (qRT-PCR) analysis to measure gene expression levels of the identified enzymes, which is generally in positive correlation with the abundance of the corresponding proteins. As shown in the Figure 5, the genes coding for enzymes in glycolysis, e.g., *pfkA*, *eno*, and *pdhB*, were slightly upregulated upon treatment with Ag^+ . More significant upregulation of genes involved in pentose phosphate pathway (*pgl* and *gnd*), ROS defense system (*ahpD*, *trxA*, *trxB*, and *bcp*) and transcription (*rpoA* and *ropB*) is noted, and the upregulation remained up to 1 hr. Thus, it is unlikely that the Ag^+ -mediated inhibition of these enzymes is induced by the downregulation of the proteins.

TEL: (852) 2859 8974
 FAX: (852) 2857 1586
 E-mail: hsun@hku.hk

Prof. Hongzhe Sun
Norman & Cecilia Professor in Bioinorganic Chemistry

DEPARTMENT OF CHEMISTRY
 THE UNIVERSITY OF HONG KONG
 POKFULAM ROAD, HONG KONG

These data and discussions are added into the revised manuscript (page 6, line 200-206 and supplementary Fig. 7).

Figure 5. Measurement of relative gene expression levels with qRT-PCR. Gene expression in *S. aureus* after treatment of 118 μM of AgNO_3 was determined by qPCR and normalized against *rrsA* and untreated control. Two-tailed t-test was used for all comparisons between two groups. Data are presented as mean \pm SEM. * $P < 0.05$, ** $P < 0.01$ and *** $P < 0.001$. NS, not significant ($P > 0.05$).

line 188: 200 micromM Ag^+ is significantly higher than the MIC50 at 120 micromM. Why was this concentration chosen?

[Response] This is an *in vitro* study to verify that Ag^+ inhibits the activity of the identified enzymes. In this study, we added Ag^+ ions into the *S. aureus* cell lysate (protein concentration of 2 mg/mL) and then measured the enzyme activity. Higher Ag^+ concentration was used to make sure that that Ag^+ ions bind to these proteins.

line 191: is the ATP abundance the intracellular ATP concentration or the ATP/ADP ratio? If I look at the SI methods it is the ATP concentration.

TEL: (852) 2859 8974
FAX: (852) 2857 1586
E-mail: hsun@hku.hk

Prof. Hongzhe Sun
Norman & Cecilia Professor in Bioinorganic Chemistry

DEPARTMENT OF CHEMISTRY
THE UNIVERSITY OF HONG KONG
POKFULAM ROAD, HONG KONG

[Response] The ATP abundance denotes the ATP concentration. We have revised as suggested.

line 201: "revolutionized". What does this mean in this sentence? Please rephrase.

[Response] Rephrased to "utilized".

line 215-216: Was the ROS the cause or an effect of cell toxicity/death.

[Response] Ag⁺ binds and disrupts a number of proteins in ROS defense system, leading to the accumulation of ROS and ultimate cell death.

line 233: "in vivo activities of Pgl and 6PGDH". Were these in-vitro activity assays with whole cells, or truly in-vivo activities?

[Response] It is the activity of Pgl and 6PGDH in the cell lysate of *S. aureus* treated with Ag⁺. We have revised accordingly.

line 249: "induced a metabolic diverting". please rephrase. Maybe: "induced the diversion of the metabolic flux from glycolysis to oxPPP"

[Response] Rephrased accordingly.

line 250: "The shift from glycolysis to oxPPP was not shown at the metabolite level? The enzymes are inhibited by Ag+ - is there independent evidence for this change of flux?"

[Response] We agree with the reviewer and rephrased this part to avoid misunderstanding.

line 253 and Fig 2H: Here are the growth curves of WT and the knockouts with and without silver presented. Please fit the lag phase, OD_{max} and maximum specific growth rates (μ_{max}) on the basis of these figures. On the basis of these curves (and biological replicates) are there significant differences in the growth rate?

The primary effect seems to be on the lag-phase. I don't think the maximum specific growth rate is that much different for any of the Ag inhibited strains.

[Response] We truly thank the reviewer's helpful comments. As suggested, we fitted the curves and obtained the growth parameters. As shown in Figure 6, the addition Ag⁺ to both *pgl* and *gnd* gene knockdown strains resulted in longer lag phases ($\lambda = 2.72$ and 3.71 h, respectively), but not significant change on the maximum specific growth rate μ_{max} , than the WT strain ($\lambda = 2.35$ h).

TEL: (852) 2859 8974
 FAX: (852) 2857 1586
 E-mail: hsun@hku.hk

Prof. Hongzhe Sun
Norman & Cecilia Professor in Bioinorganic Chemistry

DEPARTMENT OF CHEMISTRY
 THE UNIVERSITY OF HONG KONG
 POKFULAM ROAD, HONG KONG

We have incorporated the figures and descriptions into the revised manuscript (page 7, line 254 and supplementary Fig. 11).

Figure 6. Comparison of the growth parameters of WT *S. aureus* and gene knock out strains with and without treatment of Ag⁺. (A) The lag phase (λ). (B) The maximum specific growth rate (μ_{max}). (C) The maximum growth (OD_{max}). The results are shown as mean \pm SEM ($n = 3$). Two-tailed t-test was used for all comparisons between two groups. * $P < 0.05$, ** $P < 0.01$ and *** $P < 0.001$. NS, not significant ($P > 0.05$).

line 256-257: I don't follow this exactly. The knockout of two silver binding proteins makes it more sensitive to silver (apparently slower growth in the presence of Ag⁺)?

[Response] We have rephrased this part to make it clear.

line 259: the reduction of the PPP enzyme activities is evidence against the increased flux through the PPP, despite the protein upregulation.

[Response] Agreed.

line 262: Is the metabolic redirection futile, or does it not happen at all? I'm not really convinced that at the start of the Ag⁺ interaction this increased flux through the oxPPP really happens to start with. As the authors state Ag⁺ disrupts this pathway. Maybe this is a semantic discussion, the upregulation of the proteins of oxPPP is futile because the enzymes are inhibited.

[Response] We agree with the reviewer that it is not appropriate to use "metabolic redirection" since we only observed the upregulation at the enzyme level. We have rephrase this part to avoid misleading.

TEL: (852) 2859 8974
 FAX: (852) 2857 1586
 E-mail: hsun@hku.hk

Prof. Hongzhe Sun
Norman & Cecilia Professor in Bioinorganic Chemistry

DEPARTMENT OF CHEMISTRY
 THE UNIVERSITY OF HONG KONG
 POKFULAM ROAD, HONG KONG

line 268: this part is a bit confusing, as mentioned in my previous comment. Consider rephrasing this part, or provide independent evidence that the redirection of metabolic flux towards oxPPP actually happened.

[Response] Agreed and rephrased.

line 272: "induced the generation of ROS at the late stage". Or: "inhibits the suppression of ROS generation". It seems like the primary antioxidant mechanisms are disturbed by Ag⁺ which could lead to accumulation of ROS by other means.

Is ROS generation leading to this apparent change in metabolism?

[Response] Agreed and rephrased.

line 273: "the activated oxPPP contributes to the alleviation of Ag⁺ toxicity to S.aureus shortly,...". Where is the evidence to support this? In Fig 2I the activity of the oxPPP enzymes did not increase at the start.

[Response] Agreed and rephrased.

line 277: replace "home-made" by "in-house developed"

[Response] Revised accordingly.

line 293-294: Would more Ag⁺ bind if you add more Ag equivalents?

[Response] As shown in Figure 7, no more Ag⁺ binding is observed when further increasing the molar ratio of [Ag⁺]/[6PGDH] to 12.

We have incorporated the figures and descriptions into the revised manuscript (page 10, line 300 and Fig. 3A).

TEL: (852) 2859 8974
FAX: (852) 2857 1586
E-mail: hsun@hku.hk

Prof. Hongzhe Sun
Norman & Cecilia Professor in Bioinorganic Chemistry

DEPARTMENT OF CHEMISTRY
THE UNIVERSITY OF HONG KONG
POKFULAM ROAD, HONG KONG

Figure 7. MALDI-TOF mass spectra of apo-6PGDH and 6PGDH after incubation with 8 and 12 eq. of Ag⁺.

line 295: molar ratio increases to 4, but 5 eq were added. is this correct?

[Response] For the dose dependent inhibition (Fig. 3B), we used molar ratio of 5, while for the enzyme kinetics (Fig. 3C), 4 eq. was used.

line 298: "non-competitive inhibition". Ag⁺ is not an reversible inhibitor, I presume. The enzyme is chemically modified by Ag⁺, which apparently does not change the apparent K_M, indicating the substrate binding is spared, but the rate determining step is affected.

[Response] Agreed. We have added more explanation as suggested.

line 339: "Nam ring". nicotinamide?

[Response] Yes, revised accordingly.

line 391-392, Fig. 4C: The MIC values are very discrete. Identical for most passages with Ag and AgNP, or exactly 2x that value. Were the tested concentrations in large steps? Is it expected that the MIC of AgNP is exactly the same as for Ag⁺?

[Response] For Fig. 4C, the Ag⁺ or AgNP were added into 96-well plates with 2-fold serial dilution of the drugs, followed by addition of prepared bacterial inocula and incubation at 37 °C overnight, which is a standard measurement of MIC. The MIC was determined as the lowest concentration of the drug that could inhibit the growth of *S. aureus* by measuring the OD₆₀₀ via plate reader, and the measured MIC values for Ag⁺ and the AgNP are the same in this study.

line 417: how toxic are these silver levels for human cells?

[Response] The IC₅₀ values of Ag⁺ and AgNPs vary from 3 to 144 µg/mL for different mammalian cell lines (*J Nanomater*, 2015, 2015, 136765). The concentration (8 µg/mL) we used is with relatively low cytotoxicity to most cell lines.

line 458-459: again, I'm not so sure about the metabolic divergence. See comments above.

[Response] We have rephrased this part as suggested by the reviewer.

THE UNIVERSITY OF HONG KONG

TEL: (852) 2859 8974

FAX: (852) 2857 1586

E-mail: hsun@hku.hk

Prof. Hongzhe Sun

Norman & Cecilia Professor in Bioinorganic Chemistry

DEPARTMENT OF CHEMISTRY
THE UNIVERSITY OF HONG KONG
POKFULAM ROAD, HONG KONG

SI, table 2: "Protein map" is not a very informative annotation. This protein is an "MHC class II analog protein"

[Response] Agreed and revised to "MHC class II analog protein".

TEL: (852) 2859 8974
FAX: (852) 2857 1586
E-mail: hsun@hku.hk

Prof. Hongzhe Sun
Norman & Cecilia Professor in Bioinorganic Chemistry

DEPARTMENT OF CHEMISTRY
THE UNIVERSITY OF HONG KONG
POKFULAM ROAD, HONG KONG

Reviewer #3 (Remarks to the Author):

This is another very extensive comprehensive paper from this group, following up on their study of the Ag-proteome in *E. coli*, here applying their tool of LC-GE-ICP-MS to *S. aureus*.

[General Response to Reviewer #3]

We truly appreciate the referee's favorable comments and helpful suggestions on the present study as well as recognition on the major merits of our work. These suggestions are immensely useful for us to improve the manuscript.

concerns/changes/comments

- The primary discovery of the study is the identification of the *S. aureus* Ag-proteome. The authors add additional conformational validation experiments of structural biology and mutants which is good. However, I do not feel that the Ag-antibiotic synergy data fits in this study (lines 357 to 430. Although this work is good and important, it is less novel and it is lost and the findings are in this lost in the proteome study. Highlighted but lack of inclusion of specifics in the abstract. The removal of this section would allow the authors to elaborate more on their other data.

[Response] We truly appreciate the referee's kind suggestion and understand the reviewer's concern. As suggested, we elaborated more on the identification and validation of Ag⁺ targets in *S. aureus* by performing more experiments. In addition, we add a discussion section to address the link between the metalloproteomic study and the Ag-antibiotic synergy data so that it will not be overlooked by the reviewer. Indeed, the Ag-antibiotic synergy is an extension of the discovery by our metalloproteomic findings that Ag⁺ exploits multi-targeted mode of action against *S. aureus*, which suppressed the resistance frequency and endowed the sustainable efficacy of silver against *S. aureus*.

Although it has been reported that silver could enhance the antibacterial activity of antibiotics (*Sci Transl Med*, 2013, 5, 190ra181; *ChemMedChem*, 2014, 9, 1221-1230), the molecular mechanism underlying the sustainable efficacy of silver against pathogens remain unclear. Our study resolves the long-standing question about the molecular targets and mode of action of silver against *S. aureus*. Such a unique mode of action of Ag⁺ via targeting multiple pathways endows it with the sustainable efficacy against *S. aureus*. Based on the mode of action of silver, we further explore and demonstrate that Ag⁺/AgNP can potentiate the efficacy of a

TEL: (852) 2859 8974

FAX: (852) 2857 1586

E-mail: hsun@hku.hk

Prof. Hongzhe Sun

Norman & Cecilia Professor in Bioinorganic Chemistry

DEPARTMENT OF CHEMISTRY
THE UNIVERSITY OF HONG KONG
POKFULAM ROAD, HONG KONG

broad range of antibiotics, resensitize MRSA to antibiotics, and more importantly slow down the evolution of antibiotic resistance in *S. aureus*, highlighting the proof-of-principle that fundamental knowledge and molecular understanding of Ag⁺/AgNP toxicity can be translated into new antibacterial approaches.

As elucidated above, we therefore prefer keeping this part in the manuscript to extend the mechanistic discovery and reach a broader readership. We sincerely hope the reviewer could nicely follow our rational and support our decision.

-For antibiotic resistance different strains and the media can give rise to dramatic differences. Although such information is included in methods, on line 79 the strain number of the *S. aureus* used should be included as well as the media used for these experiments. This is also important as different media will lead to different proteome present in the cell. A recognition of this is important that one has not sampled the entire proteome.

[Response] We truly thank the reviewer's kind reminder and have indicated that *S. aureus Newman* was cultured in brain heart infusion (BHI) medium in the revised version.

- As some good protein chemistry and enzymology are done on some of the targets. It would be good to complement the Ag binding to RpoA Pgl and Gnd by ITC that will give affinity and validate the stoichiometries by ICP-MS.

[Response] We truly appreciate this reviewer's insightful suggestion. We performed the isothermal titration calorimetry (ITC) experiments to measure the binding affinities of Ag⁺ towards RpoA, Pgl and Gnd. For all these three proteins, ITC data exhibited two binding modes and were fitted with two-set binding models. RpoA binds 2.14 ± 0.25 and 1.36 ± 0.14 molar equivalents of Ag⁺ with apparent dissociation constants (K_d) of 0.53 ± 0.05 and 9.29 ± 1.24 μM for the first and second binding mode, respectively. For Pgl, ITC data gave rise to K_d of 7.18 ± 0.74 and 0.58 ± 0.09 μM for the first ($N_1 = 3.03 \pm 0.26$) and second ($N_2 = 5.18 \pm 0.62$) Ag⁺-binding modes respectively, while for Gnd, Ag⁺ binds to it with K_d of 1.18 ± 0.15 and 9.71 ± 0.76 μM for the first ($N_1 = 1.72 \pm 0.13$) and second ($N_2 = 3.33 \pm 0.24$) binding mode, respectively. These values generally agree well with our previous studies on the binding affinity between silver ions and proteins (at sub-micro molar level) (*Chem Sci*, 2019, 10, 7193-7199; *Chem Sci*, 2020, 11,11714) and being consistent with the stoichiometries measured by MALDI-TOF MS.

These results and descriptions have been incorporated into the revised manuscript (page 4, line 131-135 and supplementary Fig. 4).

TEL: (852) 2859 8974
 FAX: (852) 2857 1586
 E-mail: hsun@hku.hk

Prof. Hongzhe Sun
Norman & Cecilia Professor in Bioinorganic Chemistry

DEPARTMENT OF CHEMISTRY
 THE UNIVERSITY OF HONG KONG
 POKFULAM ROAD, HONG KONG

Figure 8. Isothermal titration calorimetry (ITC) results of Ag^+ binding to RpoA, Pgl and 6PGDH ($n = 3$). The titrations were carried out at 25 °C in 35 mM Tris- HNO_3 and 100 mM NaNO_3 buffer at pH 7.4. Nitrate rather than other anions is chosen to avoid potential precipitation of Ag^+ ions. The data were fitted to two-set-of-sites binding model using the Origin software. One representative of three replicates is shown. The results are shown as mean \pm SEM.

- I found it a bit vague for the location of Ag-proteins found in figure 1D as 'others' seems only two possible categories, soluble or membrane, what is meant by other location?

[Response] For the distribution of Ag^+ in different fractions of *S. aureus*, the *S. aureus* cells were fractionated into soluble, membrane and the others. Soluble proteins were collected from the supernatant of the sonicated cells, while the membrane proteins were dissolved from the pellets with PBS buffer containing 1% (w/v) SDS, and the others stand for the remaining pellet that cannot be dissolved by 1% SDS, which mainly composed by cell wall. We have incorporated the explanation into the revised manuscript.

- I was confused about the enrichment figure Fig 1 I, the text indicates Glycolytic processes were a major target yet the bar on the curve is the smallest with an intermediate FDR number. This is counter-intuitive and thus the authors should clarify in the text to explain why they feel glycolytic is more important than say nitrogen related metabolism with longer bars.

TEL: (852) 2859 8974
FAX: (852) 2857 1586
E-mail: hsun@hku.hk

Prof. Hongzhe Sun
Norman & Cecilia Professor in Bioinorganic Chemistry

DEPARTMENT OF CHEMISTRY
THE UNIVERSITY OF HONG KONG
POKFULAM ROAD, HONG KONG

[Response] For the enrichment analysis, the False Discovery Rate (FDR) is the rate that features called significant are truly null. The calculation of FDR is based on the number of identified Ag-binding proteins in a specified pathway versus total number of proteins in this pathway for *S. aureus*. Although with relative less count for glycolysis compared to that of nitrogen metabolism, but the total number proteins in *S. aureus* for glycolysis is much less than that for nitrogen metabolism, which results in the lowest FDR for glycolysis compared to other biological processes. The lowest FDR value means the lowest false positive rate and the highest significance of enrichment for this pathway, conferring the importance of this pathway.

- Section on oxidative stress and ROS, lines 197...; I think these findings are very important to define how Ag exposure leads to increased ROS. The literature is full of conflicting reports of Ag and ROS production. As Ag is not redox active it is highly unlikely that it produces ROS through Fenton like reactions. Her authors identify poisoning of the oxidative 'repair' system, which means ROS levels would increase as the repair is not functioning. NOT because Ag generates ROS. This should be stated more clearly. Also in figure 4A, it is cartooned that the Ag ions are producing ROS, which is incorrect and we must stop propagating this misinterpretation, as is nicely found from the present study, it is the poisoning of the proteins that remove ROS that leads to increased cell levels of ROS.

Also, line 201 wording is awkward; 'revolutionized by *S. aureus* to regulate the oxidative stress' should be more like 'utilized by *S. a.* to respond to oxidative stress'

[Response] We truly thank the reviewer's insightful suggestion and agree with the reviewer's comments.

-- We have carefully modified the section on "oxidation stress and ROS" as suggested. The Fig. 4A has been revised to avoid misleading. And in the discussion part, we added several sentences to clarify that the elevated ROS production in *S. aureus* upon Ag⁺ exposure is due to the binding and functional disruption of Ag⁺ ions to the proteins in the oxidative repair system.

-- For line 201, we have rephrased to "utilized by *S. aureus* to respond to oxidative stress" as suggested.

- Figure display suggestion. In Fig2 A, D, F the Ag atoms are depicted as grey balls, which is the same coloration as the enzymes in the pathway and thus difficult to notice right away. in Figure 4A the Ag atoms are red balls, be consistent and use red balls in figure 2 as well. in Figure 3 structure the Ag atoms as grey balls work ok. Also, the pathway enzymes in A are abbreviated and F full names are given. Please be consistent.

[Response] Thanks for careful reading of the figures. We have revised the figures according to the reviewer's suggestions.

TEL: (852) 2859 8974

FAX: (852) 2857 1586

E-mail: hsun@hku.hk

Prof. Hongzhe Sun

Norman & Cecilia Professor in Bioinorganic Chemistry

DEPARTMENT OF CHEMISTRY
THE UNIVERSITY OF HONG KONG
POKFULAM ROAD, HONG KONG

Reviewer #4 (Remarks to the Author):

REVIEWER COMMENTS

The manuscript entitled “Multi-target mode of action of silver against *Staphylococcus aureus* endows it with capability to combat antibiotic resistance” authored by H. Wang et al. presents a broad study applied to understand the mode of action of silver against the antibiotic-resistant bacteria *Staphylococcus aureus*.

The study uses a Liquid-Chromatography-Gel electrophoresis-Inductively Coupled Plasma Mass Spectrometry system to separate and identify 38 Ag⁺ binding proteins from different metabolic pathways. Besides, bioinformatic analysis and biochemical validations were used to demonstrate that silver targets multiple proteins, thus interfering with multiple pathways, including glycolysis, oxidative Pentose Phosphate Pathway (oxPPP), and Reactive Oxygen Species stress defense system, to exert its bactericidal effect against *S. aureus*. Furthermore, Ag⁺ ions presence was demonstrated in the crystallographic structure of 6-phosphogluconate dehydrogenase of the oxidative pentose phosphate pathway.

Overall the study provides insights into the action of silver ions on antibiotic resistant *S. aureus*. However, I am not convinced that the advances provided for some of the experiments are of sufficient quality to merit publication in *Nature communications*.

[General Response to Reviewer #4]

We truly appreciate the reviewer’s favorable comments and helpful suggestions on the present study as well as recognition on the major merits of our work. These suggestions are immensely useful for us to improve the quality of manuscript. As pointed by the other reviewers, “The study is highly interesting and provides a breakthrough by comprehensively identifying the effects of Ag on the bacterial proteome...” and given the novel discovery as well as the solid evidence to support the conclusions, we believe that this work deserves publication in *Nature Communications*.

I have some comments that the authors may want to consider to improve the quality of the manuscript:

1. The title announces the capability of silver to combat antibiotic-resistance in *Staphylococcus aureus*; however, the properties of silver and silver nanoparticles applicable to human treatments are still under

TEL: (852) 2859 8974

FAX: (852) 2857 1586

E-mail: hsun@hku.hk

Prof. Hongzhe Sun

Norman & Cecilia Professor in Bioinorganic Chemistry

DEPARTMENT OF CHEMISTRY
THE UNIVERSITY OF HONG KONG
POKFULAM ROAD, HONG KONG

investigation in laboratory and animal studies. Therefore, how can then be established that these ions combat antibiotic-resistance, in what context? How can the inhibition of human proteins be prevented?

[Response] We truly thank the reviewer's insightful comments and agree that the internal application of Ag⁺ and AgNPs to human treatments are still under investigation. In the current study, we demonstrated that the Ag⁺ and AgNPs exert sustainable antibacterial activity against *S. aureus* and the supplementation of Ag⁺ and AgNPs could restore the susceptibility of antibiotics to antibiotic-resistance *S. aureus*. The synergistic index FICI for silver and antibiotics ranges from 0.1875 to 0.375, demonstrating the high synergetic effect of Ag⁺/AgNPs and antibiotics to combat antibiotic resistance.

Despite the broad-spectrum antimicrobial activities of silver, its internal usage is restricted, which mainly due to the binding of silver to human proteins. The question raised by this referee is an excellent scientific question aroused a broad interest in the past few years. Enormous efforts have been devoted to improving the biocompatibility of Ag⁺ and reduce its toxicity via surface coating and targeted drug delivery (*ChemMedChem*, 2014, 9, 1221-1230; *Chem Rev*, 2013, 113, 4708-4754; *Chem Soc Rev*, 2014, 43, 1501-1518). More recently, we also reported a strategy of metabolic reprogramming to enhance the antibacterial activity of Ag⁺ against *E. coli* (*PLoS Biol*, 2019, 17, e3000292).

Indeed, the enhancement of the antibacterial efficacy and improvement of the biocompatibility of silver rely heavily on the understanding of its molecular mechanism of action. Particularly, mining the direct silver-targeting proteins at a proteome-wide scale is of vital importance, allowing systemic exploration of the biological pathways interrupted by silver. In this study, we focused on uncovering the molecular mechanism underlying the antibacterial activity of silver against *S. aureus*, which in turn extends the development of novel antibacterial strategy of combination of silver and antibiotics, highlighting proof-of-principle that fundamental knowledge and molecular understanding of Ag⁺/AgNP toxicity can be translated into new antibacterial approaches. We agree with the reviewer that the biocompatibility of silver warrants further investigation and added a discussion in the revised version (page 18), *i.e.*, "Nevertheless, given a large portion of essential enzymes are conserved among humans and pathogens, it is likely that silver ions and silver nanoparticles exhibit a poor selectivity to these enzymes in *S. aureus* over those in humans. Thus, despite of the enormous efforts being devoted, it remains to be a huge challenge to improve the biocompatibility and reduce the side effect of silver."

2. There is no methodology included in the main body of the manuscript. This section is a central part of the Supplementary Methods. Definitively, there should be a Methodology section with the most relevant experiments adequately explained.

TEL: (852) 2859 8974

FAX: (852) 2857 1586

E-mail: hsun@hku.hk

Prof. Hongzhe Sun

Norman & Cecilia Professor in Bioinorganic Chemistry

DEPARTMENT OF CHEMISTRY
THE UNIVERSITY OF HONG KONG
POKFULAM ROAD, HONG KONG

[Response] Great suggestion. We have added a Methods section in the revised version as suggested.

3. There is no explanation of how the crystal structures of 6D-phosphogluconate dehydrogenase, in the absence and presence of silver ions, were solved. There is a few-lines section (X-ray crystallography) in Supplementary material that, however, does not explain how the phase problem was solved for both structures (7CB6 and 7CB5). The structure's description is also incomplete; there is no indication of how many polypeptide chains exist in the asymmetric unit and if the interactions of Ag⁺ is identical. Besides, it is not explained why the apoenzyme was used for these experiments.

[Response] We truly thank the reviewer's careful reading of the manuscript and apologize for not providing enough explanations for this part. We used the structure of 6-phosphogluconate dehydrogenase from *Geobacillus stearothermophilus* (PDB: 2W8Z) as the initial model for molecular replacement to solve the apo-structure (PDB:7CB0, it will be released later). The 7CB0 structure was then used as the model for molecular replacement and searching for silver-bound and substrate-bound 6PGDH structures. WinCoot and Phenix were used for the model-building and refinement. Four polypeptide chains exist in each asymmetric unit of 6PGDH. In total, five Ag⁺ ions were observed in each 6PGDH monomer.

These explanations have been added to the main text in the revised version as suggested (page 10, line 313-319).

4. From MALDI-TOF experiments, five Ag⁺ ions bound to the enzyme 6PGDH were detected; however, no electron density map is included in Fig. 3 to judge the validity of the coordinates for ions and substrate bound to the enzyme. The author should show non-biased electron density maps for ligands.

[Response] Great advice. The non-biased electron density maps (mesh) of silver coordination sites are shown as follows, which has been incorporated into the SI as suggested by this reviewer (supplementary Fig. 12).

TEL: (852) 2859 8974
 FAX: (852) 2857 1586
 E-mail: hsun@hku.hk

Prof. Hongzhe Sun
Norman & Cecilia Professor in Bioinorganic Chemistry

DEPARTMENT OF CHEMISTRY
 THE UNIVERSITY OF HONG KONG
 POKFULAM ROAD, HONG KONG

Figure 9. Non-biased electron density maps (mesh) of Ag⁺ coordination sites in Ag-bound 6PGDH. The $2mFo-DFc$ maps (grey meshes) are contoured at 1.0σ whilst the meshes for Ag ions (orange) are contoured at 10.0σ .

5. Two preliminary validation reports are attached. The final validation reports for both structures should have been included for the revision of this manuscript.

[Response] Agreed. The final validation reports are attached as indicated by the reviewer.

6. Supplementary Table 9 (Summary of X-ray crystallography data collection and refinement) is incomplete. CC1/2 values, Rmerge or Rmeas values, redundancies, and data for the highest resolution shell are missing. B factors for all atoms should also be included.

Leave out redundant cell angles for space group 19.

[Response] We truly thank the reviewer's careful reading and apologize for not incorporating this information. We have completed the Supplementary Table 9 (as shown in Table 1) in the revised article by adding the relevant information according to the reviewer's suggestion.

Table 1. Summary of X-ray crystallography data collection and refinement statistics.

Data collection	6PG-6PGDH	Ag-6PGDH
Wavelength (Å)	0.97918	0.97919
Resolution range (Å)	80.27 - 2.64 (2.71-2.64)	80.84 - 2.54 (2.61-2.54)
Space group	$P2_12_12_1$	$P2_12_12_1$
a, b, c (Å)	90.86, 133.86, 171.32	91.92, 132.62, 169.85

TEL: (852) 2859 8974
 FAX: (852) 2857 1586
 E-mail: hsun@hku.hk

Prof. Hongzhe Sun
Norman & Cecilia Professor in Bioinorganic Chemistry

DEPARTMENT OF CHEMISTRY
 THE UNIVERSITY OF HONG KONG
 POKFULAM ROAD, HONG KONG

α , β , γ ($^{\circ}$)	90.00, 90.00, 90.00	90.00, 90.00, 90.00
Unique reflections	61985 (4549)	69042 (5067)
Rmerge(I)	0.132 (1.029)	0.113 (1.338)
Rpim(I)	0.036 (0.351)	0.032 (0.368)
CC1/2	0.999 (0.796)	0.999 (0.814)
Completeness (%)	99.78 (99.99)	99.81 (99.96)
Mean I/sigma(I)	15.3 (2.7)	17.2 (3.4)
Redundancy	13.4 (14.1)	13.5 (13.9)
Wilson B-factor	56.0	54.2
Reflections for refinement	61985 (4549)	69042 (5067)
R-work	0.205 (0.271)	0.199 (0.303)
R-free	0.246 (0.301)	0.241 (0.304)
Number of non-hydrogen atoms	14379	14529
RMSD (bonds) (\AA)	0.002	0.003
RMSD (angles) ($^{\circ}$)	0.449	0.547
Ramachandran favoured (%)	95.63	95.53
Ramachandran allowed (%)	4.15	4.37
Ramachandran outliers (%)	0.22	0.11

7. In Section “Multi-target mode of action of silver against *S. aureus* suppresses its antibiotic selection effects” and Supp. Material sections “Evaluation of mutation frequencies and selection effects of antibacterial agents” and “Combination assay”, and “Resistance studies”, Ag nanoparticles were used; however, there is no description of what kind of nanoparticles the authors used for these experiments, or how these AgNPs were prepared.

[Response] We truly appreciate the reviewer’s careful reading. In this part, PVP coated AgNP with diameter of 10 nm was used, which was purchased from nanoComposix. AgNPs were synthesized by the borohydride reduction of AgNO_3 in the presence of PVP (40 kDa) as a stabilizing agent. The AgNPs were unagglomerated and monodispersed with spherical shape as evidenced by the characterization via transmission electron microscopy (TEM), UV-Vis, and dynamic light scattering (DLS) (Figure 10 and Table 2). We have added the relevant information to the part of methods and SI (page 27 and supplementary Fig. 13).

TEL: (852) 2859 8974
 FAX: (852) 2857 1586
 E-mail: hsun@hku.hk

Prof. Hongzhe Sun
Norman & Cecilia Professor in Bioinorganic Chemistry

DEPARTMENT OF CHEMISTRY
 THE UNIVERSITY OF HONG KONG
 POKFULAM ROAD, HONG KONG

Figure 10. Images of transmission electron microscopy (TEM) of AgNPs purchased from nanoComposix. Images are adapted from the website of nanoComposix.

Table 2 Characterization of the AgNPs^a.

Number	Coating materials	Mean diameter	Size distribution (CV)	Hydrodynamic diameter (nm)	Zeta potential (mV)	λ_{\max}
N ₁₀	PVP	10.1 ± 1.8	< 25%	19	-26	390

^a Summary of the properties of AgNPs from the website of nanoComposix.

8. In some cases, there is no correspondence on what it is said in the main document with Supplementary material. For instance, it is said: “The corresponding peaks were fractionized, collected, and subjected to MALDI-TOF-MS analysis and identified through peptide mass fingerprinting after further purification by SDS-PAGE (Supplementary Fig. 3). However, the caption to Supplementary Fig. 3 is “GE-ICP-MS profile of 107Ag corresponding to Ag-415 bound RpoB at MW of 135.9 kDa.

[Response] We thank the reviewer’s careful reading. We have carefully checked the whole article to make sure the labeling of figures and tables match the corresponding contents.

Minor:

Fig. 3(H) indicates residue F238, it should say M239.

[Response] We have corrected the mislabeling of the residues in the revised manuscript.

English language revision should be performed, there are minor problems. For instance, the title: Multi-target mode of action of silver against Staphylococcus aureus endows it with the capability to combat antibiotic resistance.

THE UNIVERSITY OF HONG KONG

TEL: (852) 2859 8974

FAX: (852) 2857 1586

E-mail: hsun@hku.hk

Prof. Hongzhe Sun

Norman & Cecilia Professor in Bioinorganic Chemistry

DEPARTMENT OF CHEMISTRY
THE UNIVERSITY OF HONG KONG
POKFULAM ROAD, HONG KONG

[Response] As suggested by the reviewer, we have carefully revised the manuscript.

REVIEWER COMMENTS

Reviewer #2 (Remarks to the Author):

The authors have extensively addressed the comments and revised the manuscript accordingly. New data in support of their conclusions was added. I have no further comments and recommend publication of this manuscript.

Reviewer #4 (Remarks to the Author):

The authors have reviewed and improved the manuscript significantly. However, this reviewer still has some concerns regarding the crystal structures of the enzyme 6D-Phosphogluconate dehydrogenase.

On page 10, there is now a brief description of how the structures were determined. The section X-Ray Crystallography (lines 873-882. Page 27) should include Lines 323-323 (page 10).

There are also some problems in the description of the structures.

As sent by the authors, the validation report for the apo form is PDB-7CB6, whereas in the manuscript is indicated as PDB 7CB0.

The second validation report that was included for revision is the substrate-bound structure (PDB 7CB5).

Therefore, the authors did not include the validation report for the Ag-bound structure.

You should provide the readers with the correct three PDB entries indicated in the manuscript. This reviewer would like to examine the Ag bound structure's validation report, which was not included for revision.

New Supplementary Figure 12 now indicated that shows "Non-biased electron density maps (mesh) of Ag⁺ coordination sites in Ag-bound 6PGDH. The 2mFoDFc maps (grey meshes) are contoured at 1.0 σ whilst the meshes for Ag ions (orange) are contoured at 10.0 σ ".

The non-biased electron density maps that should be included are "omit maps" or "Polder maps", which can be generated in Phenix.

Finally, Table 1 Summary of X-ray crystallography data collection and refinement statistics Should also include the B factors for the protein, substrate, and Ag ions.

Re: “Multi-target mode of action of silver against *Staphylococcus aureus* enables its potential for combating antibiotic resistance” (NCOMMS-20-35808A)

REVIEWER COMMENTS

Reviewer #2 (Remarks to the Author):

The authors have extensively addressed the comments and revised the manuscript accordingly. New data in support of their conclusions was added. I have no further comments and recommend publication of this manuscript.

[General Response to Reviewer #2]

We truly appreciate the reviewer’s favorable comments.

Reviewer #4 (Remarks to the Author):

The authors have reviewed and improved the manuscript significantly. However, this reviewer still has some concerns regarding the crystal structures of the enzyme 6D-Phosphogluconate dehydrogenase.

[General Response to Reviewer #4]

We truly appreciate the reviewer’s favorable comments and helpful suggestions. These suggestions are immensely useful for us to improve the manuscript.

On page 10, there is now a brief description of how the structures were determined.

The section X-Ray Crystallography (lines 873-882. Page 27) should include Lines 323-323 (page 10).

[Response] We truly thank the reviewer’s careful reading of the manuscript. We have incorporated the description into the section X-Ray Crystallography (main text, page 28, line 859-867).

There are also some problems in the description of the structures.

As sent by the authors, the validation report for the apo form is PDB-7CB6, whereas in the manuscript is indicated as PDB 7CBO.

The second validation report that was included for revision is the substrate-bound structure (PDB 7CB5).

Therefore, the authors did not include the validation report for the Ag-bound structure.

You should provide the readers with the correct three PDB entries indicated in the manuscript. This reviewer would like to examine the Ag bound structure's validation report, which was not included for revision.

[Response] We truly thank the reviewer's careful reading and apologize for the mistake.

We have carefully checked the validation reports and confirm that the coordinates and structure factors for apo-, Ag-bound and 6PG-bound 6PGDH were deposited at protein databank with accessing codes of 7CB0, 7CB6, and 7CB5, respectively. The PDB-7CB6 indeed denotes the Ag-bound 6PGDH. We have informed the PDB to correct the entry title of 7CB6 (from "apo-" to "Ag-bound"). However, a new validation report is not available at current stage and for Ag-6PGDH, please refer to the validation report of 7CB6. The validation report with the correct entry title will be available when the structure is published.

New Supplementary Figure 12 now indicated that shows "Non-biased electron density maps (mesh) of Ag⁺ coordination sites in Ag-bound 6PGDH. The 2mF_o-DF_c maps (grey meshes) are contoured at 1.0σ whilst the meshes for Ag ions (orange) are contoured at 10.0σ".

The non-biased electron density maps that should be included are "omit maps" or "Polder maps", which can be generated in Phenix.

[Response] We truly thank the reviewer's insightful comments. As suggested, we have included the Polder maps for residues 138-140, 168, 211, 239-240, 357-363 and silver ions, which are generated by phenix.polder and bulk solvents are omitted, in the revised manuscript.

Supplementary Fig. 12 The 2mF_o-DF_c polder omit map of Ag⁺ coordination sites in Ag-bound 6PGDH. The electron densities for Ag (red meshes) are contoured at 7.0σ and those for the surrounding residues (green meshes) at 2.0σ.

Finally, Table 1 Summary of X-ray crystallography data collection and refinement statistics
Should also include the B factors for the protein, substrate, and Ag ions.

[Response] Agreed. The average B-factors for the protein, substrate, and Ag ions are included in the revised manuscript (SI, page 27).

REVIEWERS' COMMENTS

Reviewer #4 (Remarks to the Author):

The authors have made the corrections requested and improved the manuscript significantly. I recommend the publication of this manuscript.

I only have a suggestion: By convention, the color of the positive difference maps should be green, not red, and contoured at 3σ (Supplementary Fig. I2).

TEL: (852) 2859 8974
FAX: (852) 2857 1586
E-mail: hsun@hku.hk

Prof. Hongzhe Sun
Norman & Cecilia Professor in Bioinorganic Chemistry

DEPARTMENT OF CHEMISTRY
THE UNIVERSITY OF HONG KONG
POKFULAM ROAD, HONG KONG

Re: “Multi-target mode of action of silver against *Staphylococcus aureus* enables its potential for combating antibiotic resistance” (NCOMMS-20-35808B)

REVIEWERS' COMMENTS

Reviewer #4 (Remarks to the Author):

The authors have made the corrections requested and improved the manuscript significantly. I recommend the publication of this manuscript.

[General Response to Reviewer #4]

We truly appreciate the reviewer's favorable comments and helpful suggestions.

I only have a suggestion: By convention, the color of the positive difference maps should be green, not red, and contoured at 3σ (Supplementary Fig. I2).

[Response] We truly thank the reviewer's careful reading of the manuscript. We have revised the figure as suggested by the reviewer (SI, page 14).

Supplementary Fig. 12 The $2mF_o-DF_c$ polder omit map of Ag^+ coordination sites in Ag-bound 6PGDH. The electron densities for Ag (green meshes) are contoured at 7.0σ and those for the surrounding residues (gray meshes) at 3.0σ .